# Transcriptomic and proteomic analysis of the virulence inducing effect of ciprofloxacin on enterohemorrhagic *Escherichia coli*

Anne Cecilie Riihonen Kijewski[1], Ingun Lund Witsø[1], Arvind Y. M. Sundaram[2], Ola Brønstad Brynildsrud[3], Kristin Pettersen[4], Eirik Byrkjeflot Anonsen[5], Jan Haug Anonsen[6,7]*, Marina Elisabeth Aspholm[1]*

**1** Faculty of Veterinary Medicine, Unit for Food Safety, Norwegian University of Life Sciences, Oslo, Norway, **2** Department of Medical Genetics, Norwegian Sequencing Centre, Oslo University Hospital, Oslo, Norway, **3** Division for Infection Control, Norwegian Institute of Public Health, Oslo, Norway, **4** Norwegian Veterinary Institute, Ås, Norway, **5** Promon AS, Oslo, Norway, **6** Department of Biosciences IBV, Mass Spectrometry and Proteomics Unit, University of Oslo, Oslo, Norway, **7** Norwegian Research Centre AS, Stavanger, Norway

* marina.aspholm@nmbu.no (MEA); j.h.anonsen@ibv.uio.no (JHA)

**Data Availability Statement:** The proteomic data has been deposited in the MassIVE repository, accessible via the identifier MSV000092224. The

## Abstract

Enterohemorrhagic *E. coli* (EHEC) is considered to be the most dangerous pathotype of *E. coli*, as it causes severe conditions such as hemorrhagic colitis (HC) and hemolytic uremic syndrome (HUS). Antibiotic treatment of EHEC infections is generally not recommended since it may promote the production of the Shiga toxin (Stx) and lead to worsened symptoms. This study explores how exposure to the fluoroquinolone ciprofloxacin reorganizes the transcriptome and proteome of EHEC O157:H7 strain EDL933, with special emphasis on virulence-associated factors. As expected, exposure to ciprofloxacin caused an extensive upregulation of SOS-response- and Stx-phage proteins, including Stx. A range of other virulence-associated factors were also upregulated, including many genes encoded by the LEE-pathogenicity island, the enterohemolysin gene (*ehxA*), as well as several genes and proteins involved in LPS production. However, a large proportion of the genes and proteins (17 and 8%, respectively) whose expression was upregulated upon ciprofloxacin exposure (17 and 8%, respectively) are not functionally assigned. This indicates a knowledge gap in our understanding of mechanisms involved in EHECs response to antibiotic-induced stress. Altogether, the results contribute to better understanding of how exposure to ciprofloxacin influences the virulome of EHEC and generates a knowledge base for further studies on how EHEC responds to antibiotic-induced stress. A deeper understanding on how EHEC responds to antibiotics will facilitate development of novel and safer treatments for EHEC infections.

## Background

Enterohemorrhagic *Escherichia coli* (EHEC) is a zoonotic pathogen, which asymptomatically colonizes the bovine recto-anal junction and is transmitted to humans mainly through fecal

associated DOI for these data is doi:10.25345/C5SN01F2D. All raw RNAseq data files are publicly available in the NCBI SRA (accession number PRJNA984016).

**Funding:** The authors received no specific funding for this work.

**Competing interests:** The authors have declared that no competing interests exist.

contamination of food or water [1]. EHEC also colonizes the human colonic epithelium and its adherence to the intestinal lining is mediated by adhesive organelles such as flagella, pili, and fimbriae. In humans, EHEC infections can cause hemorrhagic colitis (HC) and hemolytic uremic syndrome (HUS), which can lead to severe sequela and death [2]. All EHEC strains express the Shiga toxin (Stx), which is regarded as the main virulence factor of this group of pathogenic *E. coli*. Stx causes intestinal tissue damage, by arresting translation [3]. However, due to the high occurrence of the globotriaosyl (Gb3) receptor, which Stx binds to, it has the most damaging effects on renal cells, endothelial cells, and neurons [3]. After Stx attaches to Gb3, it is transported into the cell, where it can cause damage resulting in apoptosis, necrosis, or inflammation [3].

Stx is encoded by both cryptic and lysogenic bacteriophages (phages). Elevated levels of the toxin are produced concomitantly with new phage particles when Stx phages enter the lytic (proliferative) cycle. After some time in the lytic cycle, the host cell bursts, resulting in the release of phage particles and large amounts of Stx into the environment [4]. It has been proposed that the released phages can infect and lysogenize susceptible intestinal *E. coli* (or other bacterial species) and turn them into Stx producers, thereby accelerating disease progression [5,6]. There are two main types of Stx, Stx1 and Stx2, which are antigenically different but have the same mode of action. Stx2 is 50–400 times more potent than Stx1 and *E. coli* that carry *stx2* are more often epidemiologically linked to severe disease than those that only carry *stx1* [7–10].

Stx prophages (phages that are incorporated in the bacterial chromosome) can enter the lytic cycle both in the absence of an external trigger (spontaneous induction) or as a response to external factors that trigger the bacterial SOS response. The SOS response is a global response to DNA damage in which the cell division is arrested, and DNA repair mechanisms are induced [11,12]. DNA damaging factors such as UV light, reactive oxygen species (ROS) and antibiotics, particularly quinolones, have been shown to be efficient inducers of the SOS response [13]. Treatment of EHEC infections with antibiotics is generally discouraged as it has been shown to increase production of Stx *in vitro* [14] and because treatment of human patients with antibiotics, in some cases, has been shown to worsen disease symptoms and increase the incidence of HUS [15–18].

O157:H7 is the most common EHEC serotype isolated from human cases [19]. Strains belonging to this serotype contain a 92 kbp pO157 plasmid, which carries 100 open reading frames (ORFs) of which many have been associated with virulence [20]. EHEC strains also carry the locus of enterocyte effacement (LEE) pathogenicity island (PAI) which encodes a type III secretion system (T3SS), various secreted effector proteins, as well as regulatory proteins. The LEE PAI also encodes the adhesin intimin and its cognate receptor (Tir). The *Tir* protein is translocated to the host cell via the T3SS and subsequently inserted into the host cell membrane, where it acts as the receptor for *intimin located on the bacterial surface* [21]. EHEC thereby promotes its own strong adhesion to the intestinal epithelium which in turn makes it easier for effector molecules to be injected into the host cell. This leads to rearrangement of the host cell actin cytoskeleton and formation of attachment and effacing (A/E) lesions [2].

Although we have some insight into why antibiotics can have a negative effect on the outcome of EHEC infections, there is limited comprehensive information on their effect on EHEC and its pathogenic potential. To increase the knowledge on how exposure to antibiotics affects EHEC, we have performed both RNA-Sequencing (RNA-Seq) and proteomic analysis (LC-MS) on the reference EHEC O157:H7 strain EDL933 (EDL933) exposed to ciprofloxacin. With this, we aim to quantitatively assess EHECs response to sub-lethal doses of the antibiotic with a special emphasis on mechanisms associated with its virulence. The results show that exposure to antibiotic treatment altered expression of 1,331 genes within 2 h. Additionally,

there were alterations in the levels of 97 and 93 proteins (P ≤ 0.05) after 3 and 12 h post-exposure, respectively. Upon antibiotic treatment, a broad spectrum of chromosomally, phage and plasmid encoded virulence factors were differentially expressed which would likely influence pathogenesis of EHEC *in vivo*. This study contributes to a greater understanding of how EHEC responds to stress induced by an antimicrobial treatment and will perhaps contribute to the knowledge base required for the development of more effective therapies against EHEC infections.

## Results and discussion

### Global changes

In the logarithmic growth phase, cultures of EHEC O157:H7 strain EDL933 were exposed to a sublethal concentration of ciprofloxacin, which was previously reported to result in a significantly increased proliferation of the Stx2 phage (BP-933W) and increased production of Stx2 [14]. As shown in Fig 1, the growth of EDL933 in the antibiotic-treated cultures started to decline relative to untreated cultures 2 h after addition of ciprofloxacin. The optical density ($OD_{600}$) of the ciprofloxacin-treated cultures reached its peak 3 h after addition of the antibiotic whereafter it decreased during the following hours. This result is in accordance with previous reports showing that the growth of EDL933 starts to decline concomitant with an increase in the phage titer 2 h after addition of ciprofloxacin to the bacterial culture [22].

### Transcriptomic analysis

Genome-wide transcriptomic analysis was performed to identify significantly differentially expressed (DE) genes (P-adj < 0.05) 2 h after ciprofloxacin was added to the bacterial culture. Out of a total of 5,370 annotated genes, exposure to ciprofloxacin resulted in DE of 24.8% (1,331) genes out of which 712 (54%) were upregulated and 619 (46%) were downregulated (Fig 2). The average upregulation of DE genes was 12-fold, and the average downregulation was -3-fold.

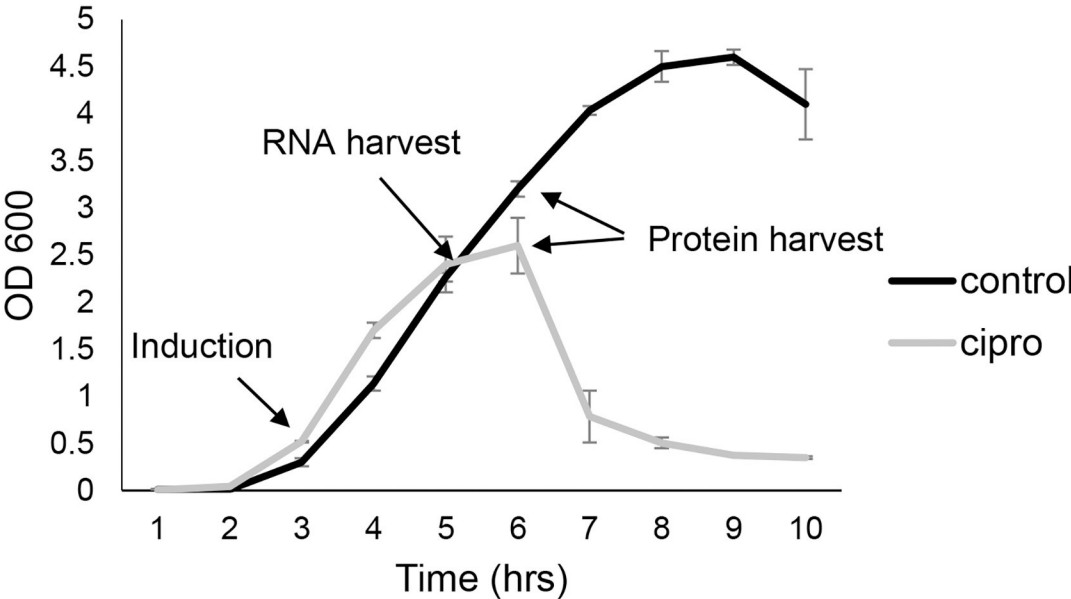

**Fig 1. The effect of ciprofloxacin (0.06 μg/mL) on the growth of EDL933 as measured by $OD_{600}$.** Results are shown as means of three independent experiments with bars showing ± standard deviation.

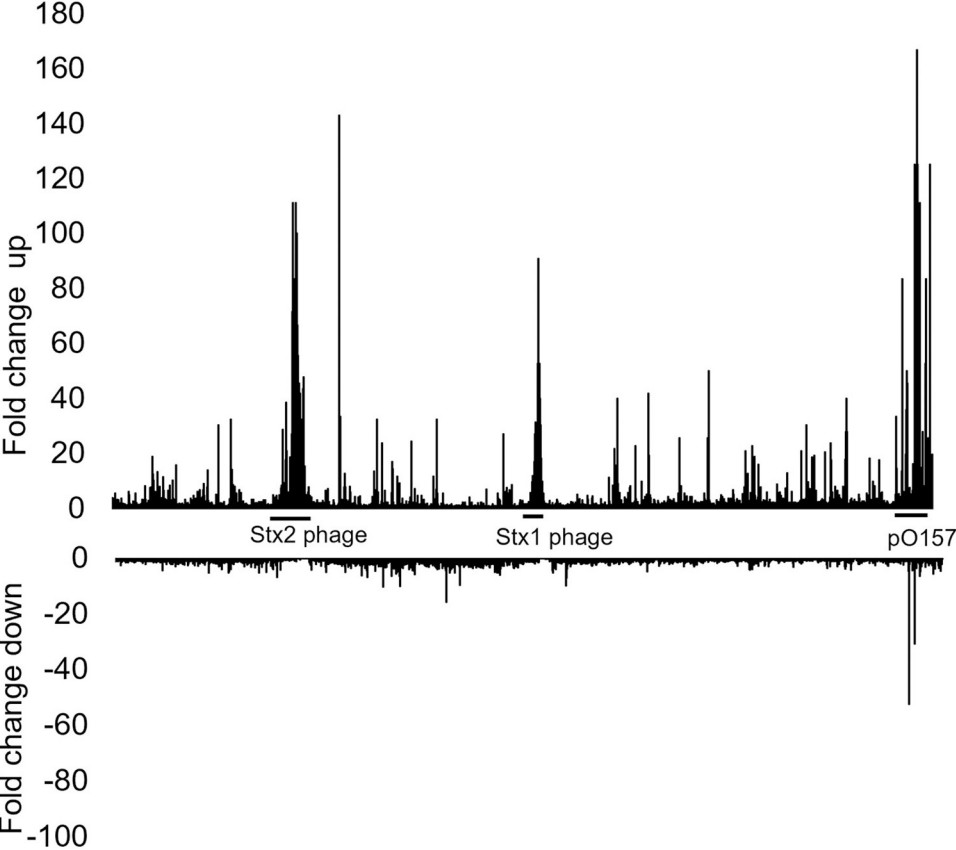

**Fig 2. A column chart showing the fold-changes in gene expression in ciprofloxacin-treated samples in comparison to untreated samples.** All 5,370 locus tags from EDL933_RS00005- EDL933_RS34060 are shown from left to the right. Regions that contain the locus tags for Stx1/2 phages (CP-933V and BP-933W) and the pO157 virulence plasmid has been marked with lines in that area.

## Proteomic analysis

Whole cell proteins were isolated from cells exposed to ciprofloxacin for 3 h or 12 h (Figs 1, 3 and S3). Cells that had not been exposed to ciprofloxacin but were harvested at the same time served as the non-exposed control. A total of 1,876 proteins (of a total of 5,730 locus tags) were identified with a posterior error probability (PEP) score lower than 0.1. A total of 43 proteins were identified as significantly less abundant, and 75 proteins were identified as significantly more abundant ($P \leq 0.05$) in the cells that were exposed to ciprofloxacin for 3 h compared to control cells (S2 Fig). For the cells harvested 12 h after addition of the antibiotic, 76 proteins were identified as significantly less abundant, and 94 proteins identified as significantly more abundant than in the control cells (S2 Fig).

## Functional enrichment of regulated genes and proteins

To get a better overview of how EHEC responds to ciprofloxacin exposure, the function of the gene products was assigned according to the tree-like hierarchical structure in the KEGG database. As illustrated by the Voronoi tree map of the transcriptome in Fig 4, only 3/5 of EDL933 genome is annotated in the KEGG database. Many of the genes that were DE following ciprofloxacin treatment encode hypothetical proteins with unknown function. Another large group of DEGs, mostly upregulated, is phage-encoded (Figs 2, 4 and S1 Table). Strain EDL933 carries one lysogenic Stx prophage BP-933W and one cryptic Stx prophage CP-933V, which encode

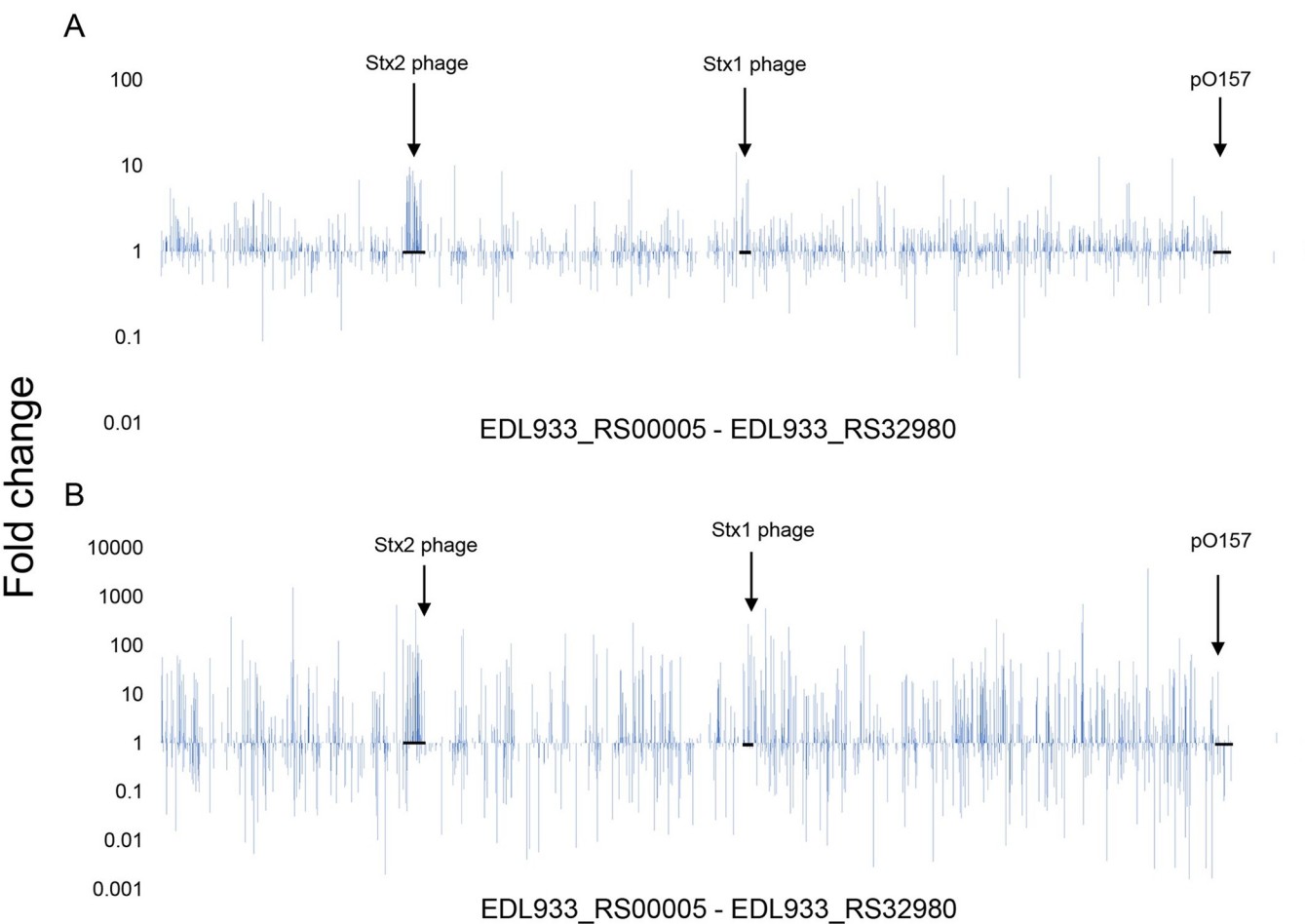

**Fig 3.** A column chart showing the fold changes in protein abundance of ciprofloxacin-treated samples at 3 h (A) and 12 h (B) in comparison to unexposed samples. All 5,370 locus tags from EDL933_RS00005- EDL933_RS34060 are shown from left to right, and proteins that were not isolated were set to 1, which also means unchanged. Columns above 1 illustrates proteins upregulated by ciprofloxacin and columns below 1 illustrates proteins that are downregulated by ciprofloxacin.

Stx2 and Stx1, respectively, as well as an additional 16 cryptic prophages [23,24]. Ciprofloxacin exposure also altered expression of many genes and proteins carried by pO157. The section showing regulation of the metabolic pathways mostly show downregulated genes.

The abundance of detected proteins was generally lower in bacteria exposed to ciprofloxacin for 12 h compared to in those exposed for only 3 h (4% lower in the control samples and 8.3% lower in the ciprofloxacin exposed samples). Despite the lower protein abundance, there was higher levels of Stx2 phage- and virulence-associated proteins in the 12 h samples compared to the 3 h samples. The proteome Voronoi tree maps (Fig 5) also show that the expression of phage proteins was higher in bacteria exposed to ciprofloxacin for 12 h compared to in those exposed to the antibiotic for 3 h. Proteins involved in metabolic pathways also showed an increased abundance in samples collected 12 h after addition of the antibiotic compared to samples collected after 3 h. This can possibly be due to the differences in growth phases between the ciprofloxacin treated cells vs. the untreated cells.

From this point forward, this paper will focus on genes and proteins related to virulence that were DE in response to ciprofloxacin exposure, with particular attention to the SOS-response, phages, the pO157 virulence plasmid, flagellar motility, adhesion and LPS synthesis.

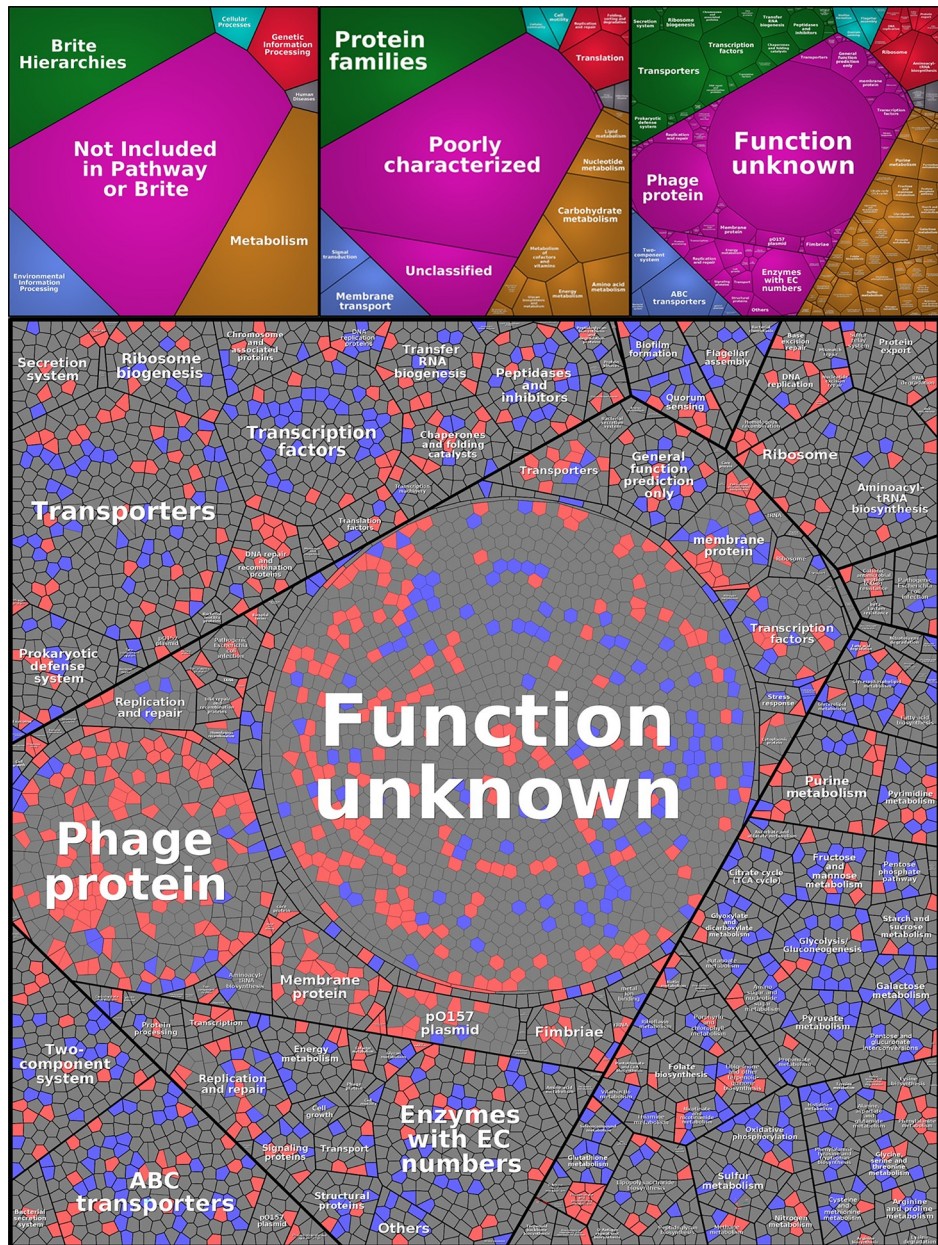

**Fig 4. The change in transcriptional pattern made by ciprofloxacin treatment of different functional categories of genes presented in a Voronoi tree map.** Red cells represent significantly upregulated genes (P-adj < 0.05), blue cells represent significantly downregulated genes (P-adj < 0.05), and gray cells represent either unchanged expression compared to uninduced control samples or DE but with P-adj > 0.05. The top panel show a general representation of the functional pathways that the genes are sorted by.

### Upregulation of the SOS response

As expected, many SOS response-associated genes were upregulated after exposure to ciprofloxacin (Table 1). Similarly, the abundance of many SOS-response associated proteins was either comparable to that of uninduced samples or increased in the samples collected 3 h after addition of ciprofloxacin (Table 1). There was an increased level of the RecA protein, which plays a key role in the in the SOS-response by stimulating the self-cleavage of LexA, leading to

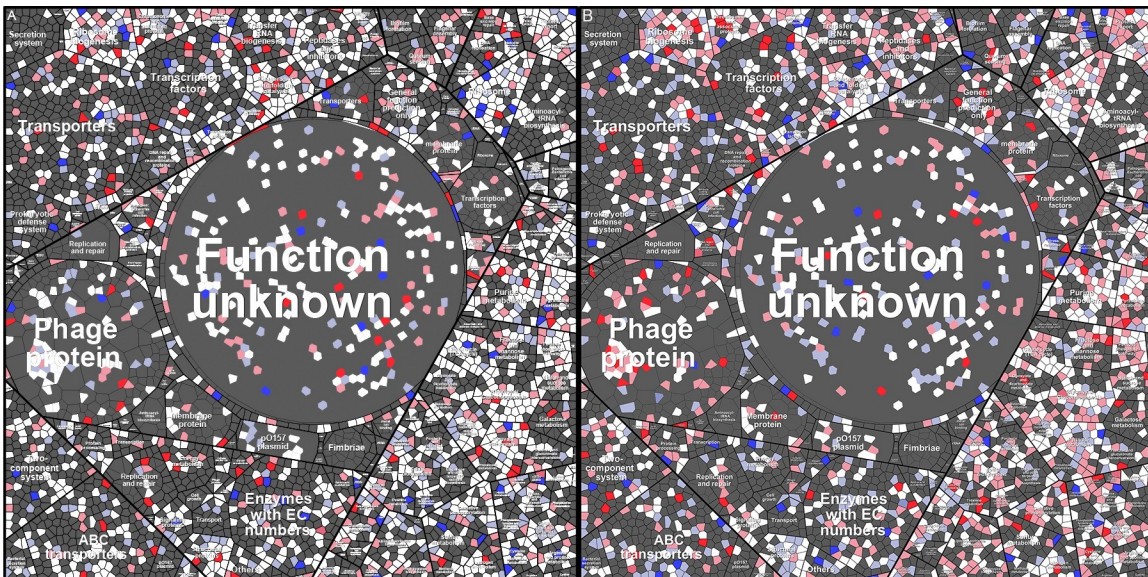

**Fig 5.** All detected proteins shown by Voronoi tree maps at 3 h (A) and 12 h (B). The fold change in protein abundance (ciprofloxacin/control) for individual proteins are indicated as follows: Sharp red = significantly increased abundance by ciprofloxacin treatment (p < 0.05), light red = increased abundance but not significant, sharp blue = significantly lower abundance (p < 0.05), light blue = lower abundance but not significant and white = proteins that were detected but found to be regulated (within parameters) P < 0.05 (Student's t-test). In the Voronoi tree maps, clustering in categories indicate functional relationships in the same way as in the top panel of Fig 3.

derepression of the SOS-regulon. Typically, genes involved in DNA repair and cell cycle arrest were also positively regulated following exposure to ciprofloxacin. In accordance with the non-septated filamentous phenotype of ciprofloxacin-treated *E. coli*/EHEC cells reported by us and others [14,25], the gene encoding the cell division inhibitor SulA was 4.6-times upregulated in ciprofloxacin-treated cells. SulA, belongs to the SOS regulon and it prevents bacterial cell division by interfering with FtsZ ring formation at the site of future cell division. There is an several reports indicating that the filamentous phenotype protects *E. coli* from being killed by innate immune cells and other unrelated insults such as antibiotics [26–28]. When the bacteria had been exposed to ciprofloxacin for 12 h, a few SOS-response associated proteins showed a much higher abundance compared to the control samples (Table 1). For example, the abundance of RecA and the DNA repair protein RuvB, was 100- and 50-times higher, respectively, in the samples containing ciprofloxacin [29].

## Upregulation of phage-associated genes and proteins

In EHEC and other bacterial pathogens, such as *Staphylococcus aureus*, the SOS response induces phage proliferation and an increase in the pathogenic potential of the host bacterium [30]. It has also been reported that *Clostridium difficile* produces more phages in response to antibiotic therapy, which was linked to elevated toxin production and hence virulence [31,32]. In this study, a total of 128 DE genes of phage origin were detected and out of these, 49 are located on BP-933W and 21 are located on CP-933V. Only 4 phage-associated genes were downregulated following addition of ciprofloxacin (-1.9- to -3.2-fold) and these are located on the cryptic prophages CP-933T and CP-933C (S1 Table). Similar changes were also seen in the proteomic data set where the majority (109) of phage proteins showed increased abundance after the addition of ciprofloxacin and 71 out of 180 detected proteins exhibited decreased levels (-1.1 — -141.6-times downregulation) (S1 Table). Three hours of ciprofloxacin exposure

**Table 1. SOS response-associated differentially expressed genes (DEG) and proteins.**

| Locus tag | Ref locus tag | Gene | Description | RNA seq Fold change 2 h | Protein Fold change 3 h | Protein Fold change 12 h |
|---|---|---|---|---|---|---|
| EDL933_RS24605 | 0 | tisB | Type I toxin-antitoxin system toxin TisB | 3.6 | --- | --- |
| EDL933_RS00330 | EDL933_0062 | polB | DNA polymerase II | 3.5 | --- | --- |
| EDL933_RS01325 | EDL933_0256 | dinJ | DNA-damage-inducible protein J | --- | 1.8 | -1.3 |
| EDL933_RS01350 | EDL933_0264 | dinB | DNA polymerase IV | 5.0 | --- | --- |
| EDL933_RS02700 | EDL933_0547 | recR | Recombination protein RecR | --- | 4.8 | -1.7 |
| EDL933_RS03175 | EDL933_0649 | hokE | Protein HokE | 3.5 | --- | --- |
| EDL933_RS03695 | EDL933_0757 | ybfE | Uncharacterized protein YbfE | --- | 1.0 | -1.3 |
| EDL933_RS04475 | EDL933_0900 | uvrB | Excinuclease ABC subunit B | --- | 1.0 | -1.1 |
| EDL933_RS04585 | EDL933_0922 | dinG | ATP-dependent helicase DinG | 2.1 | 1.1 | -1.9 |
| EDL933_RS06015 | EDL933_1225 | sulA | Cell division inhibitor SulA | 4.6 | -1.4 | -1.3 |
| EDL933_RS06545 | EDL933_1330 | yccM | 4Fe-4S binding protein | 5.7 | --- | --- |
| EDL933_RS07940 | EDL933_1637 | dinI | DNA-damage-inducible protein I | 4.2 | 10.1 | 12.0 |
| EDL933_RS09140 | EDL933_1877 | umuD | Protein UmuD | 3.9 | --- | --- |
| EDL933_RS09145 | EDL933_1878 | umuC | DNA polymerase V subunit UmuC | 2.5 | --- | --- |
| EDL933_RS13800 | EDL933_2821 | yebG | DNA damage-inducible protein | 2.2 | 3.2 | 2.5 |
| EDL933_RS13865 | EDL933_2834 | ruvB | Holliday junction DNA helicase RuvB | --- | 1.1 | 50.5 |
| EDL933_RS13870 | EDL933_2835 | ruvA | Holliday junction DNA helicase RuvA | --- | 1.8 | 1.4 |
| EDL933_RS14275 | EDL933_2918 | uvrC | Excinuclease ABC subunit C | -1.6 | --- | --- |
| EDL933_RS14280 | EDL933_2919 | uvrY | BarA-associated response regulator UvrY (GacA, SirA) | --- | 1.2 | -1.2 |
| EDL933_RS18550 | EDL933_3777 | recN | DNA repair protein RecN | 8.3 | --- | --- |
| EDL933_RS18985 | EDL933_3862 | recX | Regulatory protein RecX | 3.4 | --- | --- |
| EDL933_RS18990 | EDL933_3863 | recA | DNA recombination/repair protein RecA | 4.0 | 5.5 | 100.5 |
| EDL933_RS20100 | EDL933_4094 | recJ | Single-stranded-DNA-specific exonuclease RecJ | 1.6 | --- | --- |
| EDL933_RS24125 | EDL933_4907 | dinD | DNA damage-inducible protein D | 5.9 | -1.0 | 1.2 |
| EDL933_RS24725 | EDL933_5023 | recF | DNA replication and repair protein RecF | 2.8 | --- | --- |
| EDL933_RS25335 | EDL933_5132 | uvrD | DNA-dependent helicase | 2.8 | --- | --- |
| EDL933_RS25395 | EDL933_5144 | recQ | ATP-dependent DNA helicase RecQ | --- | -1.4 | -1.1 |
| EDL933_RS25445 | EDL933_5154 | rmuC | DNA recombination protein RmuC | 3.2 | 1.3 | 10.7 |
| EDL933_RS25595 | EDL933_5180 | polA | DNA polymerase I | --- | 1.4 | 7.7 |
| EDL933_RS26670 | EDL933_5381 | dinF | MATE family efflux transporter DinF | 5.9 | --- | --- |
| EDL933_RS26665 | EDL933_5380 | lexA | LexA repressor | 6.5 | 1.3 | -1.5 |
| EDL933_RS28195 | EDL933_5688 | symE | Hypothetical protein | 2.5 | --- | --- |
| EDL933_RS26740 | EDL933_5396 | uvrA | Excinuclease ABC subunit A | 2.3 | 1.9 | 11.8 |
| EDL933_RS26745 | EDL933_5397 | ssb1 | ssDNA-binding protein | 3.7 | 1.0 | 1.1 |

SOS response DE genes and proteins shown as fold changes between ciprofloxacin treated samples compared to the control/untreated samples. All DEG fold change values listed in the table have a statistical significance P-adj < 0.05. The table is organized chronologically by the position of the genes in the genome annotated by the locus tag. Values above 1 indicate upregulation, below 1 indicates downregulation and 1 means no change in expression level after exposure to ciprofloxacin.

resulted in an average 3-fold increased abundance of phage proteins and after 12 h, the abundance of these proteins had increased 33-fold compared to the levels found in untreated samples. Notably, both transcriptomic and proteomic data show differential regulation of many phage genes and proteins annotated as "hypothetical proteins" (S1 Table). According to proteomic data, many of them are expressed and can therefore be classified as "proteins of unknown function". Out of these, EDL933_1402, EDL933_1403, EDL933_1410, EDL933_1385 and

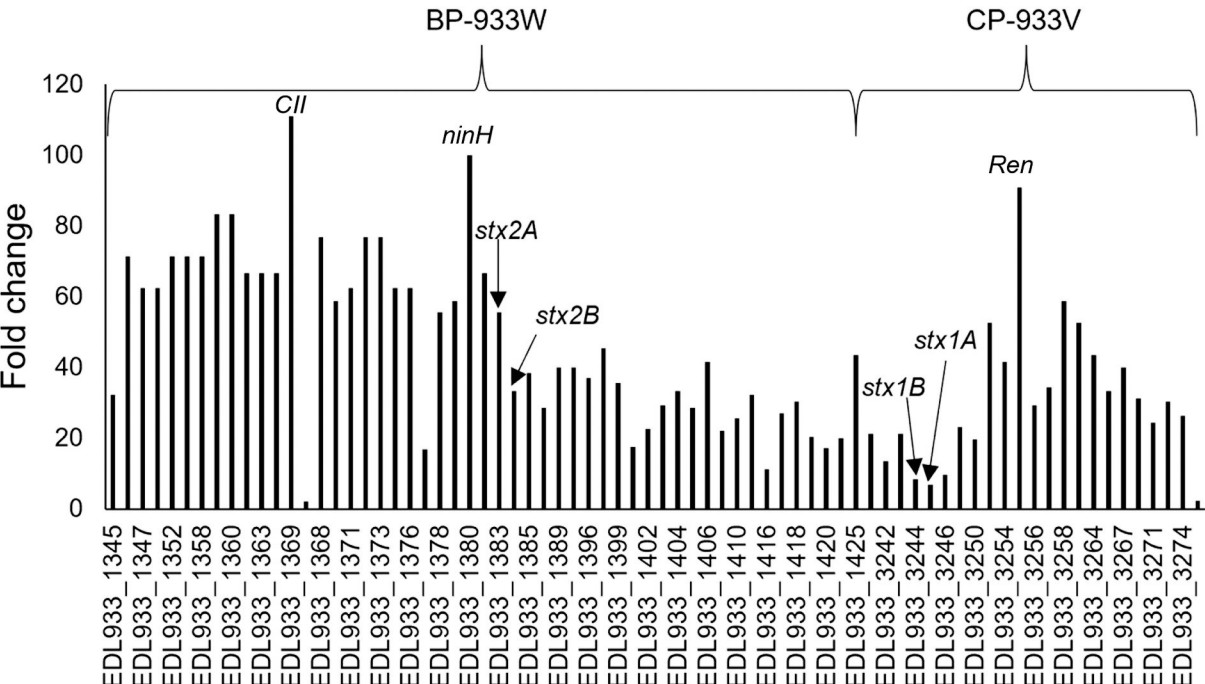

**Fig 6. A column charts showing the fold changes in gene expression in ciprofloxacin-treated samples in comparison to untreated samples in BP-933W (Stx2) and CP-933V (Stx1).** All shown data have a P-adj value < 0.05.

EDL933_1419 (S1 Table), has been shown to be overexpressed in EHEC strains that are considered highly virulent compared to EDL933 [33]. Since many of the DE ORFs encode hypothetical proteins, further research is needed to understand how the altered expression changes the behavior of the phage and the host bacterial cell (S1 Table).

The genes encoding NinH (EDL933_1380), the regulatory protein CII (EDL933_1369) of BP-933W, and exclusion protein Ren (EDL933_3255) of CP-933V exhibited the highest DE among phage genes during exposure to ciprofloxacin (highest peaks in Fig 6). Another highly upregulated phage-gene in the ciprofloxacin-treated samples (76-fold) encodes a phage protein of unknown function (EDL933_1373) that has been reported to be uniquely present in highly virulent STEC strains [34] (S1 Table). Some virulence-associated genes carried by non-Stx phages were also upregulated. The gene encoding the tail fiber protein of prophage CP-933O, EDL933_2012, was 32-fold upregulated following exposure to ciprofloxacin (S1 Table). This can possibly influence the rate of cell lysis, as this type of membrane protein (TolA) is essential for importing colicin E1 and N [35].

All Stx-encoding genes were expressed in the control samples but exposure to ciprofloxacin led to a general increase in the expression levels. Both *stx2A* and *stx2B* were highly upregulated (55-fold), while *stx1A* and *stx1B* were modestly upregulated (8.5 and 7-fold respectively). This finding aligns with a prior microarray analysis showing that the induction of genes encoding Stx1 is modest in comparison to that of genes encoding Stx2 when strain EDL933 is exposed to norfloxacin, which similar ciprofloxacin, belongs to the fluoroquinolone class of antibiotics and is an efficient inducer of the SOS-response [36].

Stx is an $AB_5$-holotoxin that consists of one A subunit that is non-covalently bound to a pentamer of five identical B subunits, co-expressed from the same operon [37]. Because of this stoichiometry and because they are encoded by the same operon, one would expect an approximately 1:5 ratio in the levels of these two subunits. Although both transcriptomic and

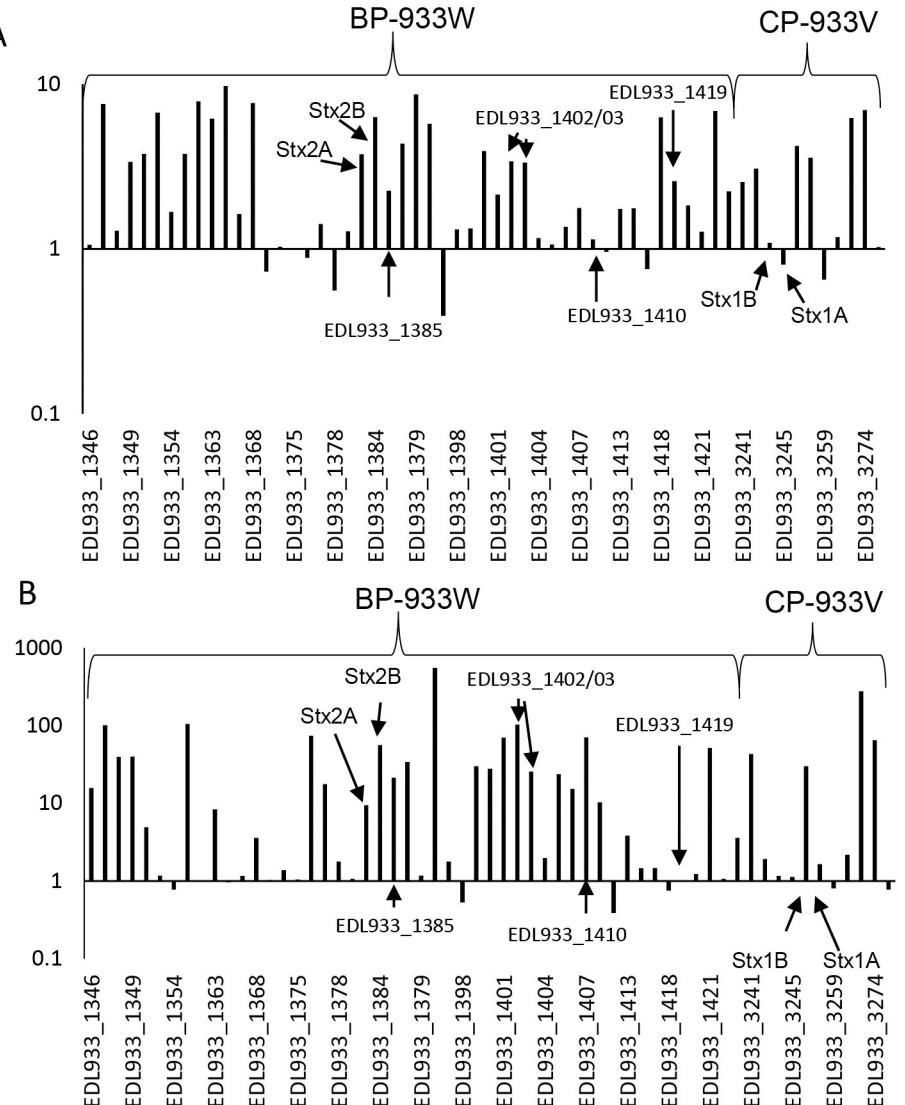

**Fig 7. A column charts showing the fold changes of protein yield in ciprofloxacin-treated samples in comparison to untreated samples in the Stx2 (BP-933W) and Stx1 (CP-933V) phages.** (A) 3 h and (B) 12 h.

proteomic data indicated increased expression of most Stx1/2 proteins, a 1:5 ratio between the A and B subunits was neither observed in the gene expression levels nor in the whole cell protein extracts at 3 h after addition of ciprofloxacin. Instead, the transcriptomic data showed 3.4-times higher transcription of *stx2A* than of *stx2B* in the ciprofloxacin-treated samples (2-times higher in uninduced samples), and the proteomic samples showed a ratio of 1.5 in Stx2B:Stx2A levels at the 3 h sampling timepoint (3.8- and 6.3- fold change for the A- and B-subunit respectively) (Fig 7). A more efficient transcription of A- compared to B-subunit genes, was also reported from the above-mentioned study where EDL933 was induced with norfloxacin [36]. After 12 h exposure to ciprofloxacin, there was a an almost tenfold increase in the levels of this toxin and the abundance of A relative to B subunits were more in accordance with the expected ratio (A- subunit = 9.4-fold B-subunit = 56-fold i.e., ratio 1:6) (Fig 7). Such an increase in toxin level is likely to increase EHEC's virulence potential in a human infection. Increased toxin production, and hence enhanced virulence in response to antibiotic

**Table 2. Differentially expressed pO157-encoded genes and proteins.**

| | | | | RNA seq | Protein | |
| | | | | Fold change | Fold change | |
| Locus tag | Ref locus tag | Gene | Description | 2 h | 3 h | 12 h |
| --- | --- | --- | --- | --- | --- | --- |
| EDL933_RS28555 | EDL933_p0001 | *finO* | Fertility inhibition protein | 1.6 | --- | --- |
| EDL933_RS28595 | EDL933_p0008 | | Hypothetical protein | --- | 1.1 | 1.3 |
| EDL933_RS28630 | EDL933_p0015 | *katP* | Catalase | --- | -1.1 | -1.7 |
| EDL933_RS28635 | EDL933_p0016 | | Hypothetical protein | --- | -1.2 | -4.5 |
| EDL933_RS28645 | EDL933_p0019 | *espP* | Per-activated serine protease autotransporter enterotoxin EspP | --- | 1.6 | 29.3 |
| EDL933_RS28675 | EDL933_p0025 | *cptA* | UPF0141 membrane protein YijP possibly required for phosphoethanolamine modification of lipopolysaccharide | --- | 1.5 | 1.4 |
| EDL933_RS28695 | EDL933_p0029 | *stcE* | Lipoprotein, ToxR-activated, TagA | --- | 1.2 | 1.1 |
| EDL933_RS28710 | EDL933_p0032 | *gspE* | Type II secretion system protein GspE | 1.3 | --- | --- |
| EDL933_RS28720 | EDL933_p0034 | *gspG* | General secretion pathway protein G | --- | 1.0 | -4.1 |
| EDL933_RS28810 | EDL933_p0051 | | Recombinase | 1.6 | --- | --- |
| EDL933_RS28825 | EDL933_p0054 | | Hypothetical protein | -2.5 | --- | --- |
| EDL933_RS28830 | EDL933_p0055 | | Hypothetical protein | -2.2 | --- | --- |
| EDL933_RS28835 | EDL933_p0056 | | Hypothetical protein | --- | -1.4 | -15.6 |
| EDL933_RS28845 | EDL933_p0058 | *ccdA* | CcdA protein (antitoxin to CcdB) | --- | -1.1 | -12.3 |
| EDL933_RS28850 | EDL933_p0059 | *ccdB* | CcdB toxin protein | --- | 2.9 | 1.1 |
| EDL933_RS28870 | EDL933_p0065 | *sopA* | Chromosome (plasmid) partitioning protein ParA | --- | -1.2 | 1.0 |
| EDL933_RS28875 | EDL933_p0066 | *sopB* | Chromosome (plasmid) partitioning protein ParB | --- | 1.1 | -1.2 |
| EDL933_RS31890 | Z_L7067 | | RepB family plasmid replication initiator protein | 10.2 | --- | --- |
| EDL933_RS28895 | EDL933_p0071 | | DNA methylase | 5.0 | --- | --- |
| EDL933_RS28915 | EDL933_p0075 | | Antirestriction protein | 33.3 | --- | --- |
| EDL933_RS28920 | EDL933_p0076 | | DUF1380 domain-containing protein | 22.7 | --- | --- |
| EDL933_RS32975 | Z_L7079 | | Hypothetical protein | 125.0 | --- | --- |
| EDL933_RS28935 | Z_L7080 | | Hypothetical protein | 9.6 | --- | --- |
| EDL933_RS28950 | Z_L7083 | | Hypothetical protein | 14.5 | --- | --- |
| EDL933_RS28940 | EDL933_p0080 | | DUF3560 domain-containing protein | 5.1 | 1.1 | -1.2 |
| EDL933_RS28960 | EDL933_p0083 | | Single-stranded DNA-binding protein | 7.4 | --- | --- |
| EDL933_RS28965 | EDL933_p0084 | | DUF905 domain-containing protein | 5.0 | --- | --- |
| EDL933_RS28970 | EDL933_p0085 | | Hypothetical protein | 3.2 | 1.1 | 2.3 |
| EDL933_RS29000 | EDL933_p0089 | | Hypothetical protein | --- | -1.2 | -1.1 |
| EDL933_RS29040 | EDL933_p0097 | | Dienelactone hydrolase-related enzyme | --- | -1.4 | 1.1 |

Plasmid pO157 DEG and proteins shown as fold changes, between ciprofloxacin-treated samples compared to the control/untreated samples. All DEG fold change values listed in the table have a statistical significance P-adj < 0.05. The table is organized chronologically by the position of the genes in the genome annotated by the locus tag. Values above 1 indicate upregulation, below 1 indicates downregulation and 1 means no change in expression level after exposure to ciprofloxacin.

treatment has also been observed in other bacterial pathogens, such as *Bacillus cereus* and *S. aureus* [38].

## Regulation of virulence associated pO157 encoded genes and proteins

The plasmid O157 (pO157) in strain EDL933 contains 100 ORFs, 16 of which were DE after addition of ciprofloxacin (Table 2). Nine of these genes are classified as hypothetical proteins or proteins with domains of unknown function. One of the hypothetical proteins is encoded by the most strongly upregulated pO157-carried gene (125-fold) (Table 2). The gene

EDL933_p0075, encoding an anti-restriction protein, was 33.3-fold upregulated in the cipro-floxacin-treated samples. Plasmid and phage encoded anti-restriction proteins protect the DNA from degradation until it has received appropriate modification or folding in the recipient cell [39]. This mechanism enhances the chance of foreign DNA to be maintained in a new bacterial host. Increased expression of the anti-restriction protein is also important for protecting the cell against DNA damaging stress.

The proteomic analysis detected 17 pO157-encoded proteins out of which eight are annotated as "proteins of unknown function" and one as "putative plasmid protein". Exposure to ciprofloxacin did not lead to any dramatic changes in the abundance of pO157-encoded proteins. The toxin CcdB showed the highest increase (4.4-fold) in protein abundance 3 h after addition of ciprofloxacin, whereas the level of its corresponding antitoxin CcdA was slightly reduced (-1.5-times) in the same samples. When such toxin-antitoxin systems are present on plasmids, they ensure that only cells that have received the plasmid survive after cell division [40]. EspP, an extracellular serine protease, was also observed at modestly higher levels (3.2-fold) in samples collected 3 h after addition of ciprofloxacin. However, after 12 h of exposure, EspP was 29-fold more abundant in antibiotic-treated samples that in untreated control samples. EspP induces macropinocytosis, which may allow Stx to cross the intestinal barrier [41]. It also cleaves the human coagulation factor V, which can cause mucosal damage leading to hemorrhage [42].

## Regulation of the LEE pathogenicity island and other host cell adherence-associated factors

A variety of verified and putative adhesins and adhesion-associated genes have been identified in EHEC. Some of those that are considered most important for virulence in EHEC are encoded by the LEE PAI. The EDL933 LEE PAI contains 41 ORFs. Treatment with ciprofloxacin did not have a large impact on the regulation of LEE-associated genes but resulted in a slight (2.1–4.0-fold) downregulation of *espK*, *espR1* and *nleA*, which encode effector proteins, and *grlA*, which encodes a transcriptional regulator (Table 3).

A total of 19 LEE-associated proteins were detected. After the bacteria had been exposed to ciprofloxacin for 3 h, many of these proteins were detected at much higher concentrations than in the untreated control samples (Table 3). This was also observed in samples collected from cells exposed to the antibiotic for 12 hours, although the increase was slightly less pronounced (average change in all LEE proteins were 9.7 in the 3 h samples and 7.5 in the 12 h samples). The increased expression of LEE-encoded proteins after addition of ciprofloxacin is in accordance with the upregulation of LEE genes and increased T3SS formation seen in EPEC cells when the SOS-response is triggered [43]. Important LEE-encoded proteins were upregulated in bacteria exposed to ciprofloxacin for both 3 and 12 h, including the needle protein EspA (59- and 35- fold, respectively), the needle pore proteins EspB and EspD (8- and 16-fold respectively, and 28- and 13-fold respectively), the translocated intimin receptor (Tir) (81- and 26-fold, respectively), the apoptosis inducing effector EspF (12- and 13-fold, respectively) and the multi effector chaperone CesT (10- and 16- fold, respectively) (Table 3) [44–46]. The strong upregulation of these proteins suggests that adhesion and type three secretion may be enhanced when the bacteria have been exposed to ciprofloxacin for longer and shorter periods. Increased T3SS expression due to ciprofloxacin treatment has also been observed in pathogenic *Pseudomonas aeruginosa* [47].

EHEC also carry non-LEE encoded genes that promote adhesion to host cells [48], and which were found to be upregulated in ciprofloxacin-treated samples. Among these were genes encoding components of type 1 fimbriae. EDL933 carries two operons encoding type 1

**Table 3. The LEE and T3SS DEGs and protein fold changes.**

| Locus tag | Ref locus tag | Gene | Description | RNA seq Fold change | Protein Fold change | |
|---|---|---|---|---|---|---|
| | | | | 2 h | 3 h | 12 h |
| EDL933_RS01340 | EDL933_0262 | *flhA* | Flagellar type III secretion system protein FlhA | 7.6 | --- | --- |
| EDL933_RS08600 | EDL933_1767 | *espK* | Type III secretion system protein | -2.6 | --- | --- |
| EDL933_RS10630 | EDL933_2172 | *espR1* | Leucine-rich repeat domain-containing protein | 2.9 | --- | --- |
| EDL933_RS11950 | EDL933_2442 | *espM1* | Putative chaperone protein | --- | -2.3 | -1.0 |
| EDL933_RS12005 | EDL933_2451 | *nleA* | Hypothetical protein | -4.0 | 2.0 | -1.1 |
| EDL933_RS19920 | EDL933_4059 | | Type III secretion inner membrane channel protein (LcrD,HrcV,EscV,SsaV) | --- | -1.3 | 1.1 |
| EDL933_RS19930 | EDL933_4061 | *escC* | Type III secretion outermembrane pore forming protein (YscC,MxiD,HrcC, InvG) | --- | -1.7 | 2.3 |
| EDL933_RS20580 | EDL933_4195 | *nleE* | Hypothetical protein | --- | 3.1 | -1.3 |
| | EDL933_4937 | *espF* | EspF | --- | 12.3 | 13.1 |
| EDL933_RS24280 | EDL933_4940 | *escF* | Type III secretion protein SsaG | --- | 1.3 | 1.2 |
| | EDL933_4942 | *espB* | Secreted protein EspB | --- | 8.1 | 16.5 |
| | EDL933_4943 | *espD* | Secreted protein EspD | --- | 28.0 | 13.9 |
| | EDL933_4944 | *espA* | EspA protein | --- | 59.2 | 35.2 |
| | EDL933_4947 | *eae* | Intimin | --- | -1.4 | -1.1 |
| EDL933_RS24320 | EDL933_4948 | *cesT* | Tir chaperone | --- | 10.2 | 16.6 |
| | EDL933_4949 | *tir* | Translocated intimin receptor Tir | --- | 81.3 | 26.3 |
| EDL933_RS24330 | 0 | *map* | Type III secretion system LEE effector Map (Rho guanine exchange factor) | 2.2 | --- | --- |
| EDL933_RS24335 | EDL933_4953 | *cesF* | ROrf10 | --- | 1.0 | 1.8 |
| EDL933_RS24350 | 0 | *escP* | Type III secretion system LEE needle length regulator EscP | 2.1 | --- | --- |
| EDL933_RS24360 | EDL933_4957 | *escN* | Type III secretion cytoplasmic ATP synthase | --- | 1.0 | 1.0 |
| EDL933_RS24365 | EDL933_4958 | *escV* | Type III secretion inner membrane channel protein (LcrD,HrcV,EscV,SsaV) | --- | -1.1 | -2.1 |
| EDL933_RS24385 | EDL933_4961 | *escJ* | Type III secretion bridge between inner and outermembrane lipoprotein (YscJ,HrcJ,EscJ, PscJ) | --- | -13.6 | 10.0 |
| EDL933_RS24405 | 0 | *grlA* | Type III secretion system LEE transcriptional regulator GrlA | 2.0 | --- | --- |
| EDL933_RS24410 | EDL933_4965 | *grlR* | Orf10 | --- | -1.2 | 1.3 |
| EDL933_RS24460 | EDL933_4975 | *ler* | Ler protein | --- | -1.4 | 8.7 |

LEE and T3SS DEGs and proteins shown as fold changes, between ciprofloxacin treated samples compared to the control/untreated samples. All DEG fold change values listed in the table have a statistical significance P-adj < 0.05. The table is organized chronologically by the position of the genes in the genome annotated by the locus tag. Values above 1 indicate upregulation, below 1 indicates downregulation and 1 means no change in expression level after exposure to ciprofloxacin.

fimbriae, one complete that contains all *fimAICDHF* (EDL933_RS10430 –EDL933_RS10455) genes needed for production of type 1 fimbriae and one incomplete (EDL933_RS27980 – EDL933_RS28025) lacking both *fimA* and *fimC*. All genes in the intact *fim* operon were 13– 17-fold upregulated in ciprofloxacin-treated samples. Nonetheless, in accordance with earlier findings, indicating the absence of functional type 1 fimbriae expression in EDL933, we were only able to detect FimB and FimC in the protein samples [49].

A total of seven non-LEE adhesion associated proteins were detected in the proteome samples but none of these showed altered abundance after addition of ciprofloxacin, except for EDL933_3181-which was 77-times more abundant in bacteria exposed to ciprofloxacin for 12 h (Table 4).

## Regulation of flagellar motility-associated genes and proteins

Swimming motility is required for EHECs to reach and colonize intestinal epithelial cells [50]. The strong antigenic properties of the flagellar filament also cause a potent immunological

**Table 4. Adherence factor DEGs and fold changes in protein abundances.**

| Locus tag | Ref locus tag | Gene | Description | RNA seq Fold change 2 h | Protein Fold change 3 h | Protein Fold change 12 h |
|---|---|---|---|---|---|---|
| EDL933_RS00110 | EDL933_0020 | *yehC* | Fimbria/pilus periplasmic chaperone | 5.4 | --- | --- |
| EDL933_RS00740 | EDL933_0145 | *yadN* | Fimbrial protein YadN | 2.4 | --- | --- |
| EDL933_RS01165 | EDL933_0224 | *tagO* | Type VI secretion system-associated protein TagO | 3.2 | --- | --- |
| EDL933_RS01395 | EDL933_0273 | | Curlin genes transcriptional activator | --- | -2.2 | -3.3 |
| EDL933_RS01855 | EDL933_0364 | *ehaA* | AidA-I adhesin-like protein | --- | 1.0 | -1.2 |
| EDL933_RS03100 | EDL933_0632 | | Type VI secretion system tip protein VgrG | 6.7 | --- | --- |
| EDL933_RS07195 | EDL933_1459 | | Fimbrial biogenesis outer membrane usher protein | 2.6 | --- | --- |
| EDL933_RS07810 | EDL933_1611 | *csgG* | Curli production assembly/transport component CsgG | -2.4 | --- | --- |
| | EDL933_1905 | | Putative adhesion and penetration protein | --- | 1.9 | -2.0 |
| EDL933_RS10430 | EDL933_2132 | *fimA* | Fimbrial protein | 5.5 | --- | --- |
| EDL933_RS10435 | EDL933_2133 | *fimC* | Fimbrial chaperone protein FimC | 16.9 | 1.5 | 1.3 |
| EDL933_RS10440 | EDL933_2134 | *fimD* | Fimbrial biogenesis outer membrane usher protein | 12.9 | --- | --- |
| EDL933_RS10445 | EDL933_2135 | *fimF* | Type 1 fimbrial adaptor subunit FimF | 14.1 | --- | --- |
| EDL933_RS10450 | EDL933_2136 | *fimG* | Type 1 fimbrial adaptor subunit FimG | 14.1 | --- | --- |
| EDL933_RS10455 | EDL933_2137 | *fimH* | Mannose-specific adhesin FimH | 14.3 | --- | --- |
| EDL933_RS15580 | EDL933_3181 | | fimbrial biogenesis outer membrane usher protein | --- | 2.5 | 77.2 |
| EDL933_RS15590 | EDL933_3183 | | Putative fimbrial-like protein | --- | -1.3 | 1.2 |
| EDL933_RS27980 | EDL933_5647 | *fimB* | Type 1 fimbriae regulatory protein FimB | 2.3 | 4.4 | -1.5 |
| EDL933_RS27985 | EDL933_5648 | *fimE* | Tyrosine recombinase | 8.2 | --- | --- |

DE adhesion genes and proteins shown as fold changes, between ciprofloxacin-treated samples compared to the control/untreated samples. All DEG fold change values listed in the table have a statistical significance P-adj < 0.05. The table is organized chronologically by the position of the genes in the genome annotated by the locus tag. Values of 1 indicate no change, values above 1 indicate upregulation by ciprofloxacin and values below 1 indicate downregulation.

reaction in the host. As a result, after a successful infection, the bacteria downregulate flagellar biosynthesis to reduce the immune response and the energy cost [51]. The downregulation of flagellar motility is also associated with an increase in self-aggregation and hence biofilm formation [50,52].

All motility related DEGs, except for *flhA* were downregulated following exposure to ciprofloxacin (-1.9 to -5.1 folds) (Table 5). Three hours after addition of ciprofloxacin, only six out of 26 motility-associated proteins were more abundant in the antibiotic-treated samples relative to the untreated samples. However, 12 h after addition of ciprofloxacin, 13 motility-associated proteins were more abundant in the samples containing the antibiotic. The most notable increases in protein abundance were seen for the methyl-accepting chemotaxis protein Trg and the flagellar hook protein FlgL that were 19- and five times upregulated, respectively. FlgL forms a structural base for the initiation of flagellar filament growth (together with FliD and FlgK), and the increased abundance of this protein can therefore be a sign of flagellar synthesis. Trg belongs to a group of proteins that acts as primary chemotaxis sensory proteins, and it has ribose and galactose as its two attractants and phenol as a repellent [53,54]. Furthermore, in the samples gathered 3 hours after addition of ciprofloxacin the abundance of the YjbJ protein was 1.7 times reduced compared to the control samples, but after 12 hours, it was 13-fold more abundant in the samples containing the antibiotic. YjbJ promotes flagellar motility, and it is likely to facilitate movement of EHEC towards the epithelial surface early in the infection process [55]. It has also been shown to repress cell adhesion and biofilm formation as well as to

**Table 5. Motility related DEGs and fold changes in protein abundance.**

| Locus tag | Ref locus tag | Gene | Description | RNA seq Fold change 2 h | Protein Fold change 3 h | Protein Fold change 12 h |
|---|---|---|---|---|---|---|
| EDL933_RS01340 | EDL933_0262 | flhA | Flagellar type III secretion system protein FlhA | 7.6 | --- | --- |
| EDL933_RS07990 | EDL933_1647 | flgM | Negative regulator of flagellin synthesis FlgM | --- | -1.3 | -1.4 |
| EDL933_RS08000 | EDL933_1650 | flgB | Flagellar basal-body rod protein FlgB | --- | -1.0 | -1.7 |
| EDL933_RS08005 | EDL933_1651 | flgC | Flagellar basal-body rod protein FlgC | --- | -1.3 | -2.3 |
| EDL933_RS08015 | EDL933_1653 | flgE | Flagellar hook protein FlgE | --- | -1.4 | 1.2 |
| EDL933_RS08020 | EDL933_1654 | FlgF | Flagellar basal-body rod protein FlgF | --- | 1.1 | 1.5 |
| EDL933_RS08025 | EDL933_1655 | flgG | Flagellar basal-body rod protein FlgG | --- | 1.3 | -1.5 |
| EDL933_RS08030 | EDL933_1656 | flgH | Flagellar L-ring protein FlgH | --- | -2.4 | 1.0 |
| EDL933_RS08045 | EDL933_1659 | flgK | Flagellar hook-associated protein FlgK | -2.1 | 1.1 | 1.2 |
| EDL933_RS08050 | EDL933_1660 | flgL | Flagellar hook-filament junction protein FlgL | -1.9 | -1.4 | 5.3 |
| EDL933_RS10890 | EDL933_2229 | trg | Methyl-accepting chemotaxis protein | -2.7 | -1.3 | 19.2 |
| EDL933_RS13970 | EDL933_2856 | cheZ | Protein phosphatase CheZ | -2.0 | -1.4 | -1.1 |
| EDL933_RS13975 | EDL933_2857 | cheY | Chemotaxis regulator—transmits chemoreceptor signals to flagelllar motor components CheY | --- | -1.9 | -1.1 |
| EDL933_RS13980 | EDL933_2858 | cheB | Chemotaxis response regulator protein-glutamate methylesterase | -1.9 | -1.5 | -1.6 |
| EDL933_RS13985 | EDL933_2859 | cheR | Protein-glutamate O-methyltransferase CheR | -2.3 | --- | --- |
| EDL933_RS13995 | EDL933_2861 | tar | Methyl-accepting chemotaxis protein II | -2.0 | -1.2 | 1.3 |
| EDL933_RS14000 | EDL933_2862 | cheW | Positive regulator of CheA protein activity (CheW) | --- | 1.1 | -2.1 |
| EDL933_RS14005 | EDL933_2863 | cheA | Signal transduction histidine kinase CheA | --- | -1.9 | 1.0 |
| EDL933_RS14010 | EDL933_2864 | motB | Motility protein B | -2.1 | --- | --- |
| EDL933_RS14020 | EDL933_2866 | flhC | Flagellar transcriptional regulator FlhC | -4.3 | --- | --- |
| EDL933_RS14025 | EDL933_2867 | flhD | Flagellar transcriptional activator FlhD | -5.1 | --- | --- |
| EDL933_RS14335 | EDL933_2931 | fliC | Flagellin FliC | -3.0 | 1.7 | 2.3 |
| EDL933_RS14340 | EDL933_2932 | fliD | Flagellar hook-associated protein 2 | -2.4 | -1.3 | 1.0 |
| EDL933_RS14345 | EDL933_2933 | fliS | Flagellar biosynthesis protein FliS | -2.4 | -1.1 | 1.3 |
| EDL933_RS14410 | EDL933_2944 | fliE | Flagellar hook-basal body complex protein FliE | --- | -2.0 | -1.2 |
| EDL933_RS14415 | EDL933_2946 | fliF | Flagellar M-ring protein FliF | --- | -1.7 | 1.2 |
| EDL933_RS14430 | EDL933_2949 | fliI | Flagellum-specific ATP synthase FliI | --- | -1.3 | -48.7 |
| EDL933_RS14450 | EDL933_2953 | fliM | Flagellar motor switch protein FliM | --- | 1.4 | -1.1 |
| EDL933_RS14455 | EDL933_2954 | fliN | Flagellar motor switch protein FliN | --- | -1.7 | -1.3 |
| EDL933_RS20840 | EDL933_4247 | qseC | Sensory histidine kinase QseC | --- | -1.8 | 1.6 |
| **EDL933_RS26675** | EDL933_5382 | yjbJ | UPF0337 protein YjbJ | --- | 2.5 | 13.2 |
| EDL933_RS28250 | EDL933_5698 | tsr | Methyl-accepting chemotaxis protein I (serine chemoreceptor protein) | --- | -1.4 | -1.1 |

Motility related DEGs and proteins shown as fold changes, between ciprofloxacin-treated samples compared to the control/untreated samples. All DEG fold change values listed in the table have a statistical significance P-adj < 0.05. The table is organized chronologically by the position of the genes in the genome annotated by the locus tag. Values of 1 indicate no change, values above 1 indicate upregulation by ciprofloxacin and values below 1 indicate downregulation.

negatively regulate expression of the curli protein CsgD in clinical EHEC O157:H7 isolates [56].

Another protein that indicates a change in the swimming pattern in response to ciprofloxacin exposure for 12 h was FliL, which was 49-times downregulated. FliL mutants are unable to "swarm" and are also slower in rotating and switching swimming direction compared to the wildtype background strain [57]. Sub-inhibitory ciprofloxacin concentrations have previously been reported to completely block swarming motility in *Salmonella enterica* (ser. Typhimurium) [58].

The gene encoding the flagellin (FliC) was 3-fold down regulated by the addition of ciprofloxacin. Nonetheless, this protein exhibited the highest abundance among all proteins in both the control samples as well as in the ciprofloxacin-treated samples collected 3 h after the antibiotic was added. In addition to being the main component of the flagellar fiber, FliC is responsible for induction of proinflammatory chemokine responses (such as IL-8), in intestinal epithelial cells [59,60].

## Regulation of genes and proteins involved in LPS synthesis

Lipopolysaccharides (LPS) are bacterial endotoxins that are major components of the Gram-negative outer membrane, which can interact with human blood platelets, cause systemic disease, and increase the risk of HUS in EHEC O157 infections [61,62]. Many genes and proteins involved in LPS synthesis were modestly DE after exposure to ciprofloxacin (Table 6). For example, genes and proteins belonging to the Lpt molecular machine involved in transport of LPS to the cell surface were slightly up or down regulated. The level of mRNA encoding the periplasmic protein LptA, which is involved in transport of LPS across the inner membrane, was 1.7-fold upregulated. The abundance of this protein was found to be 8.5-fold higher in the samples collected 3 h after addition of ciprofloxacin relative to the control samples. Similarly, LptE, which functions in the assembly of LPS at the cell surface, was 2.1-times more abundant in the cells that had been exposed to ciprofloxacin for 3 h [63]. LptG is an important inner membrane component of the Lpt transport system in *E. coli*. Without LptG, the outer membrane of *E. coli* becomes more permeable, and LPS cannot be transported to the outer leaflet of the outer membrane [64,65]. LptG showed 3-times increased abundance when the bacteria had been exposed to ciprofloxacin for 3 h and 49-times increased abundance after 12 h exposure. In accordance with the increased level of LptG, the gene encoding this protein was 1.8 times upregulated.

LpxD, which was 18- and 20-times more abundant in cells exposed to ciprofloxacin for 3 and 12 h, respectively, is an example of a protein that can impact the total level of LPS when EHEC cells are exposed to ciprofloxacin. *E. coli* cells depleted of LpxD show reduced LPS synthesis, exhibit disrupted and permeable cell walls, and show increased sensitivity to temperature and to antibiotic treatment compared to their isogenic background strain [66–68]. LpxD is a part of the lipid A biosynthesis, and more lipid A is associated with increased cytotoxicity [65].

Downregulation of some LPS genes/proteins, can also increase the production of LPS. The *lapB* gene, which encode an essential heat shock protein that plays a role in the assembly of LPS, was -2.4-fold downregulated in the ciprofloxacin-treated samples. It has previously been reported that *E. coli* cells lacking LapB show increased LPS production [69].

## Concluding remarks

This study presents transcriptomic and proteomic analyses showing how stress, induced by the fluoroquinolone antibiotic ciprofloxacin, alters the virulome of EHEC. Both the transcriptomic and the proteomic data showed that EHECs response to antibiotics is complex and involves a range of different metabolic processes and virulence-associated factors. As expected, there was increased expression of many phage-associated genes, including those encoding Stx1 and Stx2 as well as increased levels of the corresponding toxin subunits. Notably, there was also differential/upregulated expression of many other virulence-associated genes and proteins e.g., motility, T3SS and LPS (endotoxin)-synthesis. This indicates that several virulence mechanisms, besides Stx, could be involved in worsening the symptoms when EHEC infected patients are treated with antibiotics. Besides regulation of annotated genes, both the transcriptomic- and proteomic data showed altered expression of many virulence- (carried on pO157)

**Table 6. LPS biosynthesis.**

| | | | | RNA seq | Protein | |
|---|---|---|---|---|---|---|
| | | | | Fold change | Fold change | |
| Locus tag | Ref locus tag | Gene | Description | 2 h | 3 h | 12 h |
| EDL933_RS00925 | EDL933_0184 | lpxD | UDP-3-O-[3-hydroxymyristoyl] glucosamine N-acyltransferase | --- | 17.9 | 20.3 |
| EDL933_RS00935 | EDL933_0186 | lpxA | Acyl-[acyl-carrier-protein]-UDP-N- acetylglucosamine O-acyltransferase | --- | 5.8 | -1.2 |
| EDL933_RS01305 | EDL933_0252 | lpcA | Phosphoheptose isomerase 1 | --- | 14.1 | 1.0 |
| EDL933_RS03210 | EDL933_0657 | fepE | LPS O-antigen length regulator | 4.3 | --- | --- |
| EDL933_RS03485 | EDL933_0715 | lptE | LPS-assembly lipoprotein RlpB precursor (Rare lipoprotein B) | --- | 2.1 | -15.0 |
| EDL933_RS05170 | EDL933_1038 | | Phosphoethanolamine transferase EptA specific for the 1 phosphate group of core-lipid A | --- | 1.1 | 1.4 |
| EDL933_RS05785 | EDL933_1177 | msbA | Lipid A export ATP-binding/permease protein MsbA | --- | 1.6 | 7.9 |
| EDL933_RS07905 | EDL933_1630 | | Lipid A biosynthesis lauroyl acyltransferase | --- | 1.7 | -1.5 |
| EDL933_RS11795 | EDL933_2410 | lapB | LPS assembly protein B | -2.4 | --- | --- |
| EDL933_RS13840 | EDL933_2829 | lpxM | Lauroyl-Kdo(2)-lipid IV(A) myristoyltransferase | -2.1 | -1.2 | -1.3 |
| EDL933_RS15160 | EDL933_3099 | wzzB | Regulator of length of O-antigen component of lipopolysaccharide chains | --- | 1.2 | 3.1 |
| EDL933_RS15335 | EDL933_3133 | wzc | Tyrosine-protein kinase Wzc | --- | 4.8 | 1.8 |
| EDL933_RS16370 | EDL933_3341 | lpxT | Putative membrane protein | --- | 1.2 | -1.5 |
| EDL933_RS17365 | EDL933_3546 | lpxP | Lipid A biosynthesis palmitoleoyltransferase | -2.2 | --- | --- |
| EDL933_RS21730 | EDL933_4425 | kdsD | Arabinose 5-phosphate isomerase KdsD | 2 | --- | --- |
| EDL933_RS21735 | EDL933_4426 | kdsC | 3-deoxy-D-manno-octulosonate 8-phosphate phosphatase KdsC | 1.9 | --- | --- |
| EDL933_RS21740 | EDL933_4427 | lptC | Uncharacterized protein YrbK clustered with lipopolysaccharide transporters | --- | -1.2 | -1.8 |
| EDL933_RS21745 | EDL933_4428 | lptA | Lipopolysaccharide ABC transporter substrate-binding protein LptA | 1.7 | 8.5 | -1.2 |
| EDL933_RS21750 | EDL933_4429 | lptB | Lipopolysaccharide ABC transporter, ATP-binding protein LptB | --- | -5.1 | 2.8 |
| EDL933_RS23990 | EDL933_4879 | waaH | Glycosyltransferase | 2.4 | --- | --- |
| EDL933_RS24005 | EDL933_4882 | yibB | Protein YibB -involved in lipopolysaccharide biosynthesis | 2.9 | --- | --- |
| EDL933_RS24020 | EDL933_4885 | rfaC | Lipopolysaccharide heptosyltransferase RfaC | 2 | 3.5 | 4.1 |
| EDL933_RS24025 | EDL933_4886 | rfaL | O-antigen ligase RfaL | 2.3 | 4.2 | -1.2 |
| EDL933_RS24045 | EDL933_4890 | waaO | UDP-glucose:(glucosyl)lipopolysaccharide alpha-1,3-glucosyltransferase WaaO | --- | -1.6 | -1.2 |
| EDL933_RS24065 | EDL933_4894 | waaA | 3-deoxy-D-manno-octulosonic acid transferase | 2.1 | 1.5 | 2.2 |
| EDL933_RS24070 | EDL933_4895 | coaD | Phosphopantetheine adenylyltransferase | 2.8 | --- | --- |
| EDL933_RS25180 | EDL933_5105 | wzzE | Regulator of length of O-antigen component of lipopolysaccharide chains | --- | -6.9 | 1.4 |
| EDL933_RS27025 | EDL933_5457 | pmrB | Sensor protein BasS/PmrB | --- | 1.3 | 1.4 |
| EDL933_RS27035 | EDL933_5459 | eptA | Phosphoethanolamine transferase EptA specific for the 1 phosphate group of core-lipid A | --- | 1.3 | 1.0 |
| EDL933_RS27790 | EDL933_5611 | lptF | LPS export ABC transporter permease LptF | 1.7 | --- | --- |
| EDL933_RS27795 | EDL933_5612 | lptG | LPS export ABC transporter permease LptG | 1.8 | 3.0 | 49.1 |

LPS associated DEGs and proteins shown as fold changes, between ciprofloxacin treated-samples compared to the control/untreated samples. All DEG fold change values listed in the table have a statistical significance P-adj < 0.05. The table is organized chronologically by the position of the genes in the genome annotated by the locus tag. Values of 1 indicate no change, values above 1 indicate upregulation by ciprofloxacin and values below 1 indicate downregulation.

and phage-associated genes and proteins of unknown function. The potential of these proteins to contribute to the development of disease in EHEC infections remains unknown. When mapping the transcriptomic and proteomic data according to their annotated biological function, we observed a correlation between the two sets of data. However, the correlation was weaker between individual genes and proteins compared to at the functional pathway-level. The RNA was collected from cultures exposed to ciprofloxacin for 2 h while the protein extracts were harvested 3 h after addition of the antibiotic. Discrepancies could thus arise due to changes in transcript levels between the different sampling time points. For instance, the average transcript levels of SOS response-associated genes, although increased, seems lower than anticipated whereas some proteins involved in this process (Din/Yeg and UvrA) were

detected at markedly higher abundance in ciprofloxacin treated cultures (at 3 h and 12 h after induction respectively. It is also possible that some of the discrepancies between gene expression levels and protein abundance is due to post-translational modifications (PTM). PTMs could mask peptides from being identified and quantified by automatic software algorithms and could thereby influence abundance measurements. Interestingly we do detect considerable amounts of post-translationally modified proteins in the data that could indicate that there is yet another level of regulation of protein functions in EHEC. This is an unexplored area and scope for further research.

While this study provides a more holistic picture of how this EHEC responds- and adapts to antibiotic induced stress, it also highlights the large knowledge gap regarding this pathogen's genome. Further mechanistic, and *in vivo* studies are therefore needed to fully understand the pathogenic behavior of EHEC. There are several genes and proteins identified in this work that could be targeted for further studies aimed at understanding how EHEC responds to and adapts to antibiotic induced stress. Exploring these targets could potentially contribute to the development of safer and more efficient treatment regimens for EHEC infections.

## Materials and methods

### Growth experiment

EHEC strain EDL933 was grown over-night at 37˚C in 20 mL of Luria Bertani (LB) broth under agitation at 200 rpm. A volume of 20 μL of the overnight culture was transferred to 20 mL of fresh pre-warmed (37˚C) LB and grown in Erlenmeyer flasks under the same conditions as described above. The optical density at a wavelength of 600 nm ($OD_{600}$) was measured every hour and 0.06 μg/mL ciprofloxacin was added when the culture had reached 0.5 ± 0.05. The samples were covered with aluminum foil and re-incubated under the same conditions.

### Transcriptomic sample preparation

Strain EDL933 was grown in 20 mL pre-warmed LB broth in 100 mL Erlenmeyer flasks at 37˚C under agitation (200 rpm). The SOS-response was induced by adding 0.06 μg/mL ciprofloxacin to the samples when $OD_{600}$ had reached 0.5 ± 0.05. Control cultures were left uninduced. The Erlenmeyer flasks were covered with aluminum foil, to ensure dark growth conditions. After further incubation for two hours at the same conditions, 500 μL of the culture was harvested and mixed with 1 mL of RNAprotect Bacteria Reagent (Qiagen, Hilden, Germany), and stored at -80˚C until isolation of RNA.

Total RNA was extracted using the Purelink RNA mini kit (Life technologies, Carlsbad, USA) according to the manufacturer's instructions. We used the PureLink™ DNase Set (Life technologies, Carlsbad, USA) for on column removal of DNA from the samples. The quantity (A260) and purity (A260/280) of the RNA was measured using a NanoDrop 1000 spectrophotometer (Thermo Fisher Scientific, Waltham, USA) and an Agilent 2100 bioanalyzer was used to assess the quality of the RNA with the Agilent RNA 6000 nano kit (Santa Clara, California, USA). Samples with a purity of 1.90–2.10 A260/280 and with integrity over RIN 9 were sent for library preparation at Qiagen Genomic Service, Hilden, Germany.

Qiagen performed a quality control of our samples. A quantification of total amount of RNA was done on a Qubit fluorometer (Invitrogen, Carlsbad, California, USA) and the RNA integrity level was measured for each RNA sample using the Agilent TapeStation (Santa Clara, California, USA). This was done to obtain an RNA integrity value (RINe), as an indication of the quality of the RNA sample. All samples that were used in library preparations had RINe above 7.0. The library preparation and rRNA depletion was done with a combination of Bacterial FastSelect 5S/ 16S/23S (Qiagen) and TruSeq Stranded mRNA Library Prep (Illumina, San Diego, California,

USA). Quality control of the finished libraries was done at the Norwegian Center for sequencing (NGS). The NovaSeq (Illumina) sequencing was performed at NGS, with a SP1 flow cell 150 bp paired end reads. We sequenced 5 biological replicates per growth condition.

## Bioinformatic analysis

BBMap v34.56 [70] was used to remove/trim low-quality reads and adapter sequences from the raw sequence fastq files. Cleaned read pairs were mapped to the genome using hisat2 v2.1.0 [71] using genome and annotation from ENSEMBL bacteria release 47 (Escherichia_coli_o157_h7_str_edl933.ASM666v1, Escherichia_coli_o157_h7_str_edl933.ASM666v1.47.gtf). HTSeq v0.12.4 [72] was used to count the reads mapping to the genes and the differential gene expression analysis was done using DESeq2 v1.22.1 [73,74]. Raw fastq sequence data has been uploaded to NCBI SRA database under the accession number PRJNA984016.

The Voronoi tree was made with Proteomaps 2.0 at http://bionic-vis.biologie.uni-greifswald.de and the functional annotation was made with KEGG (https://www.genome.jp/entry/T00044) and BRITE hierarchies. There were 2169 genes that had no known function in KEGG or BRITE, these were looked up in the reference genome for EDL933 on NCBI: NZ_CP008957.1 and annotated accordingly. Additional information about the functional role of proteins was collected from UNIPROT and BioCyc/EcoCyc. The PCA analysis for the proteome was done with Analyse-it for Microsoft Excel (version 2.30) (S1 Fig).

## Proteomic sample preparation

An overnight culture of strain EDL933 [23] was grown for 15–16 h in 30 mL LB broth at 37˚C under agitation at 200 rpm. A volume of 100 μL of the overnight culture was added to 100 mL of pre-warmed (37˚C) LB broth. The bacteria were then cultured as described above until they reached $OD_{600}$ 0.5 ± 0.05 (exponential growth phase). Ciprofloxacin (0.06 μg/mL) was added to five of the ten bacterial cultures and the other five were left untreated. After 3 and 12 h incubation under dark conditions, 50 mL of the cultures were harvested into 50 mL falcon tubes (Corning™) and the bacteria were pelleted by centrifugation (4,000 $x$ $g$, 10 min, 4˚C). The pellets were solved in 0.8 mL 50 mM, ice cold, triethylammonium bicarbonate buffer (TEAB, Sigma Aldrich), and transferred to 1.5 mL microcentrifuge tubes and kept on ice. The bacteria where then killed/inactivated by placing them in a water bath holding 80˚C for 15 min. The samples were immediately cooled down on ice, and 0.8 mL of ice-cold buffer A (50 mM TEAB, 2% sodium deoxycholate (SDC), and one tablet proteinase inhibitor (PI)) (cOmplete Tablets, Mini EDTA-free, EASY pack, Roche) with two tablets of PI was added. The samples and buffer were mixed by pipetting and vortexing, and subsequently pelleted by centrifugation (2˚C, 10,000 x $g$, 5 min) and stored at -80˚C until further processing. The samples were thawed in room tempered water [75] and sonicated on wet ice at >60kHz for 10 x 10 s, with a 30 s pause between sonication sessions to avoid overheating of samples [76]. The bacterial cells were then pelleted by centrifugation at 20,000 x $g$, for 30 min at 2˚C. The supernatants were collected, resuspended in buffer A and centrifuged again two times to remove remaining cell debris. After removing the cell debris, the supernatants from both spin cycles were combined and the protein concentration was estimated by measuring A280 nm on a NanoDrop 1000 spectrophotometer (Thermo Fisher Scientific, Waltham, USA). For each individual sample, 20 μg protein was processed further.

## Protein reduction, alkylation, and tryptic digestion

The method for protein reduction and alkylation was modified from the method used in Kijewski et al, 2020 [14]. The protein solution containing 20 μg protein was adjusted to a

volume of 100 μL by the addition of 50 mM TEAB. A volume of 2 μL of 100 μM dithiothreitol (DTT, Sigma Aldrich) solved in 50 μM TEAB was added to the samples followed by incubation at 37˚C for 1 h, to reduce disulfide bonds. The samples were cooled to room temperature, and 8 μL of 100 mM iodoacetamide (IAA, Sigma Aldrich) solved in 50 mM TEAB was added for alkylation of the proteins' free sulfhydryl groups on cysteine residues. The samples were then incubated in the dark for 1 h. To quench excess IAA, 4 μL of the same DTT solution was added to the samples followed by addition of 33 μL or of Sequencing Grade Modified Trypsin (100 μg/mL, Promega). The samples were then incubated for 2 h at 37˚C for tryptic digestion of the proteins [14].

## Acid precipitation of SDC

The removal of the LC-MS incompatible SDC and remaining lipids was done with an acid precipitation; a method modified from Scheerlink *et al*, 2015 [77]. The samples were adjusted to 2% v/v trifluoroacetic acid (TFA), vortexed thoroughly, and incubated at room temperature for 5 min. Next, the samples were centrifuged at 21,130 x g for 10 min, and the supernatant was harvested and vacuum dried (Savant Spd 121P speed vac concentrator, Thermo Scientific, Waltham, Massachusetts, USA). The samples were re-hydrated with 147 μL 50 mM TEAB, and the acid precipitation was repeated to ensure optimal removal of SDC. The samples were then desalted with Pierce™ C18 spin tips (Thermo Scientific, Waltham, Massachusetts, USA) according to the manufacturer's instructions, and vacuum dried before storage.

## LC-MS/MS analysis

The peptide samples were resuspended in 0.1% formic acid (FA) and analyzed on two different LC-MS systems. 4 (of 5) of the 12 h samples were analyzed on an Ultimate 3000 nano-HPLC (Dionex, Sunnyvale, CA, USA) coupled to an LTQ-Orbitrap XL (OXL) mass spectrometer (MS) (ThermoElectron, Bremen, Germany). Whereas all 3 h samples and 3 of the 12 h samples (the final sample from each sample group) were analyzed using an Ultimate 3000 nano-UHPLC system (Dionex, Sunnyvale, CA, USA) connected to a QExactive (QEx) MS (Thermo-Electron, Bremen, Germany) equipped with a nano electrospray ion source. For liquid chromatography separation, on both HPLC's, an Acclaim PepMap 100 column (C18.3 μm beads, 100 Å, 75 μm inner diameter, 50 cm) (Dionex, Sunnyvale CA, USA) was used. For the OXL Ultimate 3000 nano-UHPLC system, a flow rate of 300 nL/min was employed with a solvent gradient of 3–5% B for 10 min, 5–60% for 103 min to 90% B for 2 min and maintaining that for 5 min then back to 3% B for 1 min. For the Ultimate 3000 nano-UHPLC system A flow rate of 300 nL/min was employed with a solvent gradient of 3–55% B for 53 min, to 96% B for 2 min and maintaining that for 5 min then back to 3% B in 3 min. Solvent A was 0.1% formic acid and solvent B was 0.1% FA/90% acetonitrile.

For the LTQ-Orbitrap XL (O XL), the MS was operated in the data-dependent mode (DDA) to automatically switch between MS and MS/MS acquisition. The survey full scan was acquired at a resolution, R = 60,000 (at m/z 400) from m/z 190 to m/z 2000 with an AGC target of 5.0 x $10^5$ and maximum ion accumulation time of 200 ms. The seven most intense ions (threshold 500) from the full scan survey were selected for fragmentation by collision-induced dissociation (CID) with a normalized collisional energy (NCE) of 35. MS/MS targeted ions were dynamically excluded for 180 s with an isolation window of m/z = 2 without offset. The lock mass option was enabled in MS mode for internal recalibration during the analysis (at m/z 445.12003).

For the Qexactive, the MS was operated in the DDA mode. Survey full scan MS spectra (from m/z 200 to 2000) were acquired with the resolution R = 70,000 at m/z 200, after

accumulation to a target of $1.0 \times 10^6$ The maximum allowed ion accumulation times were 100 ms. The method allowed sequential isolation of up to ten of the most intense ions (intensity threshold $1.7 \times 10^4$), for fragmentation using higher-energy collision induced dissociation (HCD) at a target value of 10,000 charges and a resolution R = 17,500 with NCE 28. Target ions already selected for MS2 were dynamically excluded for 60 s. The isolation window was $m/z$ = 2 without offset. The maximum allowed ion accumulation for the MS/MS spectrum was 60 ms. For accurate mass measurements, the lock mass option was enabled in MS mode for internal recalibration during the analysis.

## Database search and label-free quantitation

Data were acquired using Xcalibur v2.5.5 and raw files were processed. Database searches were performed against the *E. coli* O157:H7 strain EDL933 (NCBI: taxid155864; 7913 unique entries) and the proteome discoverer (PD) common contaminants list, with the PD v 2.4 software (ThermoScientific, Whaltham, Massachusetts, USA). The following parameters were used: digestion enzyme, trypsin; maximum missed cleavage, 2; minimum peptide length 4; parent ion error tolerance, 10.0 ppm; fragment ion mass error tolerance, 0.04 Da; and fixed modifications, carbamidomethylation of cysteines. Oxidation of methionine and acetylation of the N-terminus were specified as variable modifications and the maximum number of PTMs was set to 2. Peptide-spectrum matches was assessed with percolator with false discovery rate (FDR) target set at 0.01 (strict) and 0.05 (relaxed). Generated protein lists were manually curated, with low FDR proteins, proteins with single (low score) peptides, and contaminants removed.

For Label-free quantitation (LFQ) in PD v 2.4 software the following strategy was employed for the 12-h samples. Protein abundances and LFQ in PD were determined based on summed abundances of connected peptide groups and the protein fold changes from the pairwise peptide ratios. Although OXL commonly record precursor ions at a sufficient accuracy, a limiting factor of OXL based LFQ is the amount of identified peptides. To mitigate the lack of peptide ID, a single sample from each group was a-priori selected (last in each group) for QEx analysis focused on peptide ID. PD utilize the sample with greatest peptide ID in each quantitative group as a template and align the chromatographic RT by fitting a regression curve between the samples based on matching features (identified peptides with high confidence PSM only) and subsequently use this curve to match unidentified precursor ions in samples to identified peptides in the template sample based on the RT adjustment and within the specified timeframe (Max shift 20 min). Thus, the high confidence peptide identification Qex analysis offers a template to increase the amount of quantitative data in OXL analysis, provided there is sufficient accuracy and sensitivity on MS1 precursor level. Subsequently, protein abundances and normalizations were calculated by sample groups with a single QEx sample and 4 OXL samples as independent biological replicates. In effect, this attempts to emphasize the QEx data over the OXL data.

For all samples, label-free quantitation (LFQ) in PD v 2.4 software was based on the intensity values of identified unique and razor peptides with chromatographic RT alignment (max shift at 20 min). Protein abundances were based on summed abundances of connected peptide groups and the protein fold changes from the pairwise peptide ratios with missing values imputed with low abundance resampling and normalization by total peptide amount, excluding modified species. Maximum fold change was set to 100 and the p-values calculated by a background-based t-test. The data was uploaded to the MassIVE mass spectrometry data repository: doi:10.25345/C5SN01F2D.

## Supporting information

**S1 Table. Bacteriophage encoded DEGs and changes in protein abundance, shown as fold changes, isolated at 3 and 12 h post induction by ciprofloxacin.** Values above 1 illustrates upregulation in fold change, values below 1 illustrates downregulation and 1 means unchanged. All DEGs listed have P < 0.05. The sorting into different phages were done according to the EHEC O157:H7 strain EDL933 reference genome NCBI: AE005174.2. (XLSX)

**S1 Fig. Principle component analysis (PCA) plots.** RNA samples (A) and protein samples (B). C = Ciprofloxacin U = Uninduced.
(XLSX)

**S2 Fig.** Volcano plot depicting the proteomes at 3 h (A) and 12 h (B) post induction with ciprofloxacin. The most upregulated proteins are located towards the right (red), the most downregulated are towards the left (blue). The most statistically relevant are towards the top. All above log10 = 1.3 is below P-adj < 0.05. The grey markers represent proteins that have a log2 fold change up to 0.25 (up or down), which only amounts to approximately 19% change.
(XLSX)

**S3 Fig.**
(XLSX)

## Acknowledgments

The success of this project was made possible by the contributions of Kristin O'Sullivan, Toril Lindbäck, Yngvild Wasteson, and Trine L'Abée-Lund.

## Author Contributions

**Conceptualization:** Anne Cecilie Riihonen Kijewski, Ingun Lund Witsø, Arvind Y. M. Sundaram, Kristin Pettersen, Jan Haug Anonsen, Marina Elisabeth Aspholm.

**Data curation:** Anne Cecilie Riihonen Kijewski, Ingun Lund Witsø, Arvind Y. M. Sundaram, Jan Haug Anonsen.

**Formal analysis:** Anne Cecilie Riihonen Kijewski, Arvind Y. M. Sundaram, Jan Haug Anonsen.

**Funding acquisition:** Marina Elisabeth Aspholm.

**Investigation:** Anne Cecilie Riihonen Kijewski, Ingun Lund Witsø, Jan Haug Anonsen, Marina Elisabeth Aspholm.

**Methodology:** Anne Cecilie Riihonen Kijewski, Ingun Lund Witsø, Arvind Y. M. Sundaram, Ola Brønstad Brynildsrud, Kristin Pettersen, Jan Haug Anonsen, Marina Elisabeth Aspholm.

**Project administration:** Anne Cecilie Riihonen Kijewski, Marina Elisabeth Aspholm.

**Resources:** Anne Cecilie Riihonen Kijewski, Marina Elisabeth Aspholm.

**Software:** Anne Cecilie Riihonen Kijewski, Arvind Y. M. Sundaram, Ola Brønstad Brynildsrud, Eirik Byrkjeflot Anonsen, Jan Haug Anonsen.

**Supervision:** Marina Elisabeth Aspholm.

**Validation:** Anne Cecilie Riihonen Kijewski, Jan Haug Anonsen.

**Visualization:** Anne Cecilie Riihonen Kijewski, Ola Brønstad Brynildsrud, Eirik Byrkjeflot Anonsen, Jan Haug Anonsen.

**Writing – original draft:** Anne Cecilie Riihonen Kijewski, Marina Elisabeth Aspholm.

**Writing – review & editing:** Anne Cecilie Riihonen Kijewski, Ingun Lund Witsø, Arvind Y. M. Sundaram, Jan Haug Anonsen, Marina Elisabeth Aspholm.

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
