## [Decision Letter · Decision Letter 0]

7 Sep 2023

PONE-D-23-24337Transcriptomic and proteomic analysis of the virulence inducing effect of ciprofloxacin on enterohemorrhagic Escherichia coliPLOS ONE

Dear Dr. Aspholm,

Thank you for submitting your manuscript to PLOS ONE. After careful consideration, we feel that it has merit but does not fully meet PLOS ONE’s publication criteria as it currently stands. Therefore, we invite you to submit a revised version of the manuscript that addresses the points raised during the review process.

We look forward to receiving your revised manuscript.

Kind regards,

Fernando Navarro-Garcia

Academic Editor

PLOS ONE

Journal Requirements:

Reviewers' comments:

Reviewer's Responses to Questions

**Comments to the Author**

1. Is the manuscript technically sound, and do the data support the conclusions?

Reviewer #1: No

Reviewer #2: Yes

2. Has the statistical analysis been performed appropriately and rigorously? 

Reviewer #1: Yes

Reviewer #2: Yes

3. Have the authors made all data underlying the findings in their manuscript fully available?

Reviewer #1: Yes

Reviewer #2: Yes

4. Is the manuscript presented in an intelligible fashion and written in standard English?

Reviewer #1: Yes

Reviewer #2: No

5. Review Comments to the Author

Reviewer #1: PONE-D-23-24337

Transcriptomic and proteomic analysis of the virulence inducing effect of ciprofloxacin on enterohemorrhagic Escherichia coli.

Kijewski, A., et al.

This manuscript provides an -omic analysis (transcriptome and proteome) of the prototype EHEC strain EDL933 in presence of ciprofloxacin. The large set of data demonstrate that in addition to Stx induction and SOS response, other gene products with diverse functions in the bacteria are affected. However, the analysis of the data and the conclusions are highly speculative because there is no experimental evidence that several of the proteins affected are involved in the virulence process, including the hypervirulence phenotype described in the manuscript. It is evident that antibiotic treatment increases the possibility of HUS due to increase lysis and release of Stx toxin, but the data presented here does not demonstrate that changes in e.g., motility, LPS, LEE, “hypervirulence genes” is associated with complications observed during infection. As presented, the paper requires additional validation of the results and/or modification of the discussion and conclusions. Here are some comments for the authors’ consideration:

Major points:

1. Page 4 lanes 76-82, is there a need to describe the LEE-related functions instead of focusing the introduction more into the mechanisms of Stx regulation in the context of antibiotic treatment and development of HUS.

2. In the introduction, it might be also useful to describe how the development of HUS is associated with Stx production and dissemination to target organs and whether there is any evidence that addition of antibiotics makes EHEC more virulent, activating other virulence factors, instead of just a scenario where bacteria is lysed and Stx is released to the milieu and could reach the bloodstream.

3. Page 5 lane 109 and Fig 1, why was CFU/mL not determined instead or in addition to OD600?

4. The -omics analysis done in the manuscript is based on the number of genes up- or down-regulated, with particularly emphasis on those genes that are multiple fold elevated. However, the authors do not offer an explanation in page 6 lanes 116-118 and fig 2, that the fold change and magnitude is significantly higher in up-regulated genes than in those down-regulated.

5. Page 7 lanes 146-150, this text does not describe the information from Fig 4. It must be revised.

6. Page 7 lane 156, what is “higher virulence potential”? Do you want to refer to more Stx toxins produced, or more virulence-associated genes expressed? For an EHEC infection to be effective, attachment must occur because without close interaction with the intestinal cell, virulence increased is not guaranty.

7. Figures 4 and 5, the authors need to provide in the text a better explanation about the patterns observed in those figures; otherwise, the information becomes useless as a visual aid.

8. Many parts of the discussion are speculative, and the authors do not have any data to support many of their statements, except for the fact that the genes or proteins are upregulated or downregulated at 2 time points. The authors need to reduce the speculative nature of the discussion or instead, provide additional data supporting some of the statements. For example, in page 9 lanes 190-193, do the EHEC cells treated with ciprofloxacin increase their filamentous phenotype? Page 12, lanes 244-247 “to produce a seemingly inutile toxin” there is no evidence in the results that this is the case. Perhaps this result must be validated with other method as they did for the norfloxacin study.

9.

10. Page 11 lane 231-234, the authors mentioned that a prior study with norfloxacin showed similar results to the current study, what are the similarities observed between the 2 studies?

11. Page 16, lanes 320-323 can the authors corroborate whether A/E lesion formation is increased under ciprofloxacin treatment, as suggested by their results.

12. Page 17 lanes 342-352, there is no evidence that type 1 fimbriae is expressed in EHEC EDL933 or that this up regulation results in a type 1 fimbriae mediated increased in adhesion.

Minor points:

1. Page 2, lane 24, ”the most dangerous pathotype” based on what information, EHEC is the define as the most dangerous? ETEC and EPEC produce more cases and deaths than EHEC.

2. Page 3, lane 47, isn’t pili and fimbriae the same surface structures?

3. Page 4 lane 73 and elsewhere “kb” instead of “kbp”

4. Figure legends should go at the end of the manuscript to prevent disruption of the text to be evaluated.

5. Page 5 Lane 103, use “h” instead of “hrs”

6. Page 7 lane 145, what is “DE”?

7. Page 11 lane 218, EDL933_3235 is missing from Fig 6.

8. Page 12 lanes 254-255, all the genes described in this section are not depicted in Fig 7.

9. Table 3, the genes listed in the table should not be capitalized (e.g., NleE, EspF, EscF, EspB, etc)

10. All the tables’ footnotes should be in a different font size, so they do not look like part of the manuscript text.

11. Page 17, lane 355, StcC has never been associated with adhesion and it is also missing from Table 4.

12. Pages 20 and 21, the LPS section is pure speculation and there is no additional data in the paper validating their statements. This section, like others, must be reduced and speculative statements, eliminated.

13. Pages 23-24, the hypervirulence section is also speculative. First, there is not such thing as hypervirulence genes. The fact that these genes were upregulated in the clade 8 strain does not guaranty that they are involved in the hypervirulence observed with that strain. Because EDL933 is not a hypervirulent strain, there is no evidence that ciprofloxacin is inducing a hypervirulent phenotype.

14. The same concerns as stated in the previous 2 points, are applicable to flagellar motility genes (e.g., Page 26 lanes 496-500.

Reviewer #2: In this work, Kijewski and colleagues provide new insight on the global response of enterohemorrhagic E. coli to ciprofloxacin exposure. By using transcriptomic and proteomic approaches, the authors demonstrate that ciprofloxacin treatment differentially modulates EHEC gene expression, leading to an increase in the expression and production of several virulence factors such as stx1 and stx2. The conclusions are supported by the experimental data shown in the manuscript. I believe that the authors’ findings advance our knowledge of how EHEC fine-tunes its response to environmental clues, particularly after antibiotic treatment. However, significant efforts are necessary to improve the manuscript’s clarity and conciseness.

Major comments:

Please provide clearer subheaders in the results section. It is difficult to understand the main findings with the current ones.

In order to better visualize the changes in gene expression between the different experimental conditions, I suggest complementing Figure 3 with a clustergram or a volcano plot that displays the proteins that are significantly upregulated or downregulated.

Although I acknowledge that ciprofloxacin treatment causes various genes to be differentially expressed, many of which encode proteins with unknown functions, I believe that the authors could have provided a better emphasis on the known differentially expressed virulence factors and their potential role in the pathogenesis of EHEC during antibiotic treatment.

I understand the authors' intention of merging the Results and Discussion sections, but the current layout is challenging to comprehend and lacks sufficient discussion on research done in other bacterial pathogens, like Acinetobacter baumannii, etc., and their relevance to their study.

It is important that the proteins/genes mentioned in the text are clearly labeled in the corresponding figures.

Many LEE-associated proteins are downregulated at 3 h post-antibiotic treatment and strongly upregulated at 12 h. Could you speculate how this could benefit the pathogen? Could it contribute to enhanced virulence?

There appears to be a significant increase in EDL933_1388 protein levels, but not at the transcriptional level. How do you explain this? Do you have any insight on how this protein might function during antibiotic-induced stress?

Minor comments:

Line 45: …through faecal contamination of food or water

Line 109: EHEC O157:H7 str. EDL933

Line 110: Results are shown as…

Line 125: Fig. 3 is not presented in the paragraph

Line 183: remove duplication ‘in the”

Line 193: It seems that this sentence is part of the results presented by this study, but it also has a reference (29).

Line 225: it would be helpful to label the three genes described in the text in Figure 6, similarly to stx1/2.

Figure. 8: I'm not certain if the figure contains valuable information. Could you please include more details in the legend for clarification?

Authors should provide more detailed explanations in the Material and Methods section on how protein ratios relate to proteomic data. Alternatively, they could refer to it as "fold change" for clarity.

The text contains numerous inconsistencies in the terminology used. I recommend consolidating the terms throughout the text.

6. PLOS authors have the option to publish the peer review history of their article (what does this mean?). If published, this will include your full peer review and any attached files.

Reviewer #1: No

Reviewer #2: No

---

## [Author Response · Author response to Decision Letter 0]

23 Sep 2023

Reviewer #1: PONE-D-23-24337

Transcriptomic and proteomic analysis of the virulence inducing effect of ciprofloxacin on enterohemorrhagic Escherichia coli.

Kijewski, A., et al.

This manuscript provides an -omic analysis (transcriptome and proteome) of the prototype EHEC strain EDL933 in presence of ciprofloxacin. The large set of data demonstrate that in addition to Stx induction and SOS response, other gene products with diverse functions in the bacteria are affected. However, the analysis of the data and the conclusions are highly speculative because there is no experimental evidence that several of the proteins affected are involved in the virulence process, including the hypervirulence phenotype described in the manuscript. It is evident that antibiotic treatment increases the possibility of HUS due to increase lysis and release of Stx toxin, but the data presented here does not demonstrate that changes in e.g., motility, LPS, LEE, “hypervirulence genes” is associated with complications observed during infection. As presented, the paper requires additional validation of the results and/or modification of the discussion and conclusions. Here are some comments for the authors’ consideration:

Major points:

Major points:

1. Page 4 lanes 76-82, is there a need to describe the LEE-related functions instead of focusing the introduction more into the mechanisms of Stx regulation in the context of antibiotic treatment and development of HUS.

Author response:

First, we would like to thank the reviewer(s) for turning our attention to the LEE proteins, because without it we would not have discovered a mistake that had been made in the first submission. As it turns out, there had been a shift in columns in the excel sheet for alle the LEE data, and they were therefore incorrect. That has now been corrected, and we are very grateful to the reviewer for guiding us in that direction. 

We believe that a publication on EHEC virulence should discuss the LEE encoded type three secretion system, as it is one of two major virulence factors in EHEC. Furthermore, we found 7 LEE genes that were differentially expressed during exposure to ciproflocxacin. Exposure to ciprofloxacin also resulted in increased abundances of important type three secretion proteins at both 3 and 12 hours after induction with ciprofloxacin (EspA, Tir, EspB, CesT, EspD, EspF) which we mean has the potential to influence type three secretion, and therefore potentially A/E lesion formation and disease development. 

2. In the introduction, it might be also useful to describe how the development of HUS is associated with Stx production and dissemination to target organs and whether there is any evidence that addition of antibiotics makes EHEC more virulent, activating other virulence factors, instead of just a scenario where bacteria is lysed and Stx is released to the milieu and could reach the bloodstream.

Author response:

We have added a sentence in line 53, which clarifies that it is Stx’s damage to kidney cells (and endothelial and neurons) that cause HUS. We also believe that lines 68-77 adequately describe how antibiotic exposure has been demonstrated to promote Stx production in vitro and to exacerbate disease development during EHEC infection. Ciprofloxacin has been reported to increase expression of virulence factors in other bacteria [1, 2]. However, there is still a knowledge gap regarding whether antibiotic treatment can lead to higher expression of other virulence-associated factors than Stx in EHEC, which the current article helps to filling. 

3. Page 5 lane 109 and Fig 1, why was CFU/mL not determined instead or in addition to OD600?

Author response:

 We chose to only measure growth by CFU/mL due to the filamentous morphology EHEC exhibits during ciprofloxacin treatment, observed in Kijewski et al, 2020 [3]. Filamentation occurs when bacterial cells proliferate without forming septa between the mother and the daughter cells. This means that one filament can contain many nuclei (10 – 50 or even more), but during plate count only form one colony. If the filament is introduced into a favorable medium again, it will septate and there will be many more CFUs than what was measured before. Measuring OD therefore seemed more correct, as the elongated cell would influence the absorbance approximately as much as the same amount of septated cells.

4. The -omics analysis done in the manuscript is based on the number of genes up- or down-regulated, with particularly emphasis on those genes that are multiple fold elevated. However, the authors do not offer an explanation in page 6 lanes 116-118 and fig 2, that the fold change and magnitude is significantly higher in up-regulated genes than in those down-regulated. 

Author response:

Thank you for pointing this out. We have now added a sentence describing the average fold changes of up- and down-regulated genes. 

5. Page 7 lanes 146-150, this text does not describe the information from Fig 4. It must be revised.

Author response:

Thank you. We have now edited the text to better refer to Fig 4.

6. Page 7 lane 156, what is “higher virulence potential”? Do you want to refer to more Stx toxins produced, or more virulence-associated genes expressed? For an EHEC infection to be effective, attachment must occur because without close interaction with the intestinal cell, virulence increased is not guaranty.

Author response:

We agree with the reviewer that EHEC infection requires attachment to host cells in order to cause severe disease; nevertheless, there is scientific evidence that many other bacterial factors also influence the outcome of an EHEC infection. This section has been rewritten to make it clearer. The heightened LEE protein expression also suggest that adhesion can be heightened by ciprofloxacin treatment. 

7. Figures 4 and 5, the authors need to provide in the text a better explanation about the patterns observed in those figures; otherwise, the information becomes useless as a visual aid.

Author response:

We have now added to and edited the sections figures 4 and 5 belong to, to explain clearer what can be observed in the figures (line 150-171).

8. Many parts of the discussion are speculative, and the authors do not have any data to support many of their statements, except for the fact that the genes or proteins are upregulated or downregulated at 2 time points. The authors need to reduce the speculative nature of the discussion or instead, provide additional data supporting some of the statements. For example, in page 9 lanes 190-193, do the EHEC cells treated with ciprofloxacin increase their filamentous phenotype?

Author response:

The paper's aim was to provide a global view on how exposure to ciprofloxacin influences the virulome of EHEC O157:H7 strain EDL933. Unfortunately, we do not presently have the funding resources or a biosafety level 3 laboratory to conduct additional functional or mechanistic studies on the effects of ciprofloxacin on EHEC EDL933. However, we believe, and hope that you agree, that the existing findings are still valuable and relevant to the scientific community, even without further experiments. Parts of the results are supported by experimental evidence from our previous paper (Kijewski et al. 2020 [3]). Furthermore, our previous (Kijewski et al. 2020 [3]) as well as other studies have shown that exposure to ciprofloxacin triggers filamentation of EHEC cells (page 9, line 200-203). 

The number of speculative sentences in the revised manuscript has been reduced. In the discussion, we have analyzed the data in the context of existing literature, suggested further research and highlighted the study's significance and possible impact. 

9. Page 12, lanes 244-247 “to produce a seemingly inutile toxin” there is no evidence in the results that this is the case. Perhaps this result must be validated with other method as they did for the norfloxacin study.

Author response:

 Stx consists of one A subunit and five B subunits. The B subunits are necessary for attachment of the toxin to the Gb3 receptor [4]. If no B subunits are produced, the A subunit (which is the toxic part) cannot enter kidney, endothelial or neurons via the Gb3 receptor and cause HUS. 

To clarify this, we have revised this sentence L 268-272, page 13.

10. Page 11 lane 231-234, the authors mentioned that a prior study with norfloxacin showed similar results to the current study, what are the similarities observed between the 2 studies? 

Author response:

We agree with the reviewer that this must be clarified. The sentence is rephrased in the revised manuscript. It now reads: This finding aligns with a prior microarray analysis indicating that the induction of genes encoding Stx1 is modest in comparison to that of genes encoding Stx2 when strain EDL933 is exposed to norfloxacin, which similar ciprofloxacin, belongs to the fluoroquinolone class of antibiotics and is an efficient inducer of the SOS-response (31). 

11. Page 16, lanes 320-323 can the authors corroborate whether A/E lesion formation is increased under ciprofloxacin treatment, as suggested by their results.

Author response:

As the reviewer suggests, the strong upregulation of important T3SS proteins after 3 and 12 hours of ciprofloxacin induction can be interpreted to affect development of A/E lesion formation in the situation of a human infection (line 352-354, page 17).

12. Page 17 lanes 342-352, there is no evidence that type 1 fimbriae is expressed in EHEC EDL933 or that this up regulation results in a type 1 fimbriae mediated increased in adhesion.

Author response:

We agree with the reviewer and the lack of evidence that functional type 1 fimbriae are expressed is noted in the text (line 362-364). We have rephrased the sentence that this pertains to for clarity.

Minor points:

1. Page 2, lane 24,”the most dangerous pathotype” based on what information, EHEC is the define as the most dangerous? ETEC and EPEC produce more cases and deaths than EHEC. 

Author response:

While ETEC and EPEC cause more deaths worldwide, EHEC is still considered the most dangerous pathotype, due to the severity of the symptoms it causes, and due to the difficulty in treating it. Both ETEC and EPEC can be treated with antibiotics, and rehydration in the hospital/supportive treatment, eliminates the lethality of EPEC/ETEC. The severe sequela/complications one can get from an EHEC infection, such as kidney damage and failure, IBS or other gastrointestinal issues, neurological damage, and joint issues like reactive arthritis, are also a part of why it is considered to be the most dangerous E. coli pathotype [5]. Furthermore, EHEC also has a much lower infectious dose (100 CFU or even lower) compared to ETEC and EPEC (≥108 CFU) [6, 7]. 

2. Page 3, lane 47, isn’t pili and fimbriae the same surface structures?

Author response:

 These structures are sometimes used interchangeably in literature but the term pili is typically used for longer and thicker filaments that can serve more specialized functions, such as conjugation, transport of proteins, in motility and to facilitate attachment. The term fimbriae is on the other hand often used for shorter, thinner, and more numerous filaments that primarily are dedicated to attachment to biotic and abiotic surfaces and colonization.

3. Page 4 lane 73 and elsewhere “kb” instead of “kbp”

Author response:

Thanks for pointing this out. This has been corrected throughout the entire manuscript. 

4. Figure legends should go at the end of the manuscript to prevent disruption of the text to be evaluated. 

Author response:

 We followed the PLOS one author guidelines that says: “Figure captions must be inserted in the text of the manuscript, immediately following the paragraph in which the figure is first cited (read order). Do not include captions as part of the figure files themselves or submit them in a separate document.”

5. Page 5 Lane 103, use “h” instead of “hrs” 

Author response:

We have changed it to “h” throughout the document.

6. Page 7 lane 145, what is “DE”? 

Author response:

This is an abbreviation for “differentially expressed”, which we introduced in line 115.

7. Page 11 lane 218, EDL933_3235 is missing from Fig 6.

Author response:

 Gene EDL933_3235 is not mentioned in lane 218. If the reviewer is referring to EDL933_3255, this is the high column between EDL933_3254 and EDL933_3256. Due to figure sizing, only every second gene/column is marked with the gene-name.

8. Page 12 lanes 254-255, all the genes described in this section are not depicted in Fig 7.

Author response:

Due to figure sizing, only every other gene/column is marked with the gene name.

9. Table 3, the genes listed in the table should not be capitalized (e.g., NleE, EspF, EscF, EspB, etc) 

Author response:

Thank you! This is corrected in the revised manuscript.

10. All the tables’ footnotes should be in a different font size, so they do not look like part of the manuscript text. 

Author response:

The PLOS one authors guideline states that authors should use a standard font size, which is 12 pt for Times New Roman. The guidelines do not state that there should be a difference in font size between table legends and the general text. 

11. Page 17, lane 355, StcC has never been associated with adhesion and it is also missing from Table 4. 

Author response:

Thank you for pointing this out. This protein was mislabeled. The protein has now been called by its locus tag in the text (lane 355), the description in table 4 has been changed to the description on EDL933_3181 listed in the reference genome (NZ_CP008957.1).

12. Pages 20 and 21, the LPS section is pure speculation and there is no additional data in the paper validating their statements. This section, like others, must be reduced and speculative statements, eliminated.

Author response:

According to reviewers’ recommendation, we have removed some of the more speculative statements from this section.

13. Pages 23-24, the hypervirulence section is also speculative. First, there is not such thing as hypervirulence genes. The fact that these genes were upregulated in the clade 8 strain does not guaranty that they are involved in the hypervirulence observed with that strain. Because EDL933 is not a hypervirulent strain, there is no evidence that ciprofloxacin is inducing a hypervirulent phenotype.

Author response:

We have changed any reference of “hypervirulence genes/proteins” to “hypervirulence associated/related genes/proteins”.

Secondly, the layout of this paper combines the Results and the Discussion and some “speculation” is therefore warranted. As ciprofloxacin treatment of EHEC infections has made patients sicker, the discovery that some genes and proteins previously shown to be overexpressed in hypervirulent EHEC strains have their expression levels increased by ciprofloxacin treatment could prove to be an important finding. It is unknown why treatment with ciprofloxacin (or other antibiotics) has made patients with EHEC infection sicker, although it is thought that increased Stx production is to blame. We do not disagree with that, but the current findings suggest that other virulence factors may also be involved in the increased disease severity caused by antibiotic treatment. As we state, more study is needed to better understand EHEC virulence features and how antimicrobial therapies can affect clinical outcomes. 

14. The same concerns as stated in the previous 2 points, are applicable to flagellar motility genes (e.g., Page 26 lanes 496-500.

Author response:

As in our answer to question 13, we argue that due to the combination of the Results and the Discussion part, speculating/hypothesizing on what consequences the downregulation of genes of a known virulence factor (motility), is not only justified but necessary. Presenting these results and their potential importance to the scientific community is important, as they can be a foundation for further research.

Reviewer #2: In this work, Kijewski and colleagues provide new insight on the global response of enterohemorrhagic E. coli to ciprofloxacin exposure. By using transcriptomic and proteomic approaches, the authors demonstrate that ciproflox

---

## [Decision Letter · Decision Letter 1]

9 Nov 2023

PONE-D-23-24337R1Transcriptomic and proteomic analysis of the virulence inducing effect of ciprofloxacin on enterohemorrhagic Escherichia coliPLOS ONE

Dear Dr. Aspholm,

Thank you for submitting your manuscript to PLOS ONE. After careful consideration, we feel that it has merit but does not fully meet PLOS ONE’s publication criteria as it currently stands. Therefore, we invite you to submit a revised version of the manuscript that addresses the points raised during the review process.

We look forward to receiving your revised manuscript.

Kind regards,

Fernando Navarro-Garcia

Academic Editor

PLOS ONE

Journal Requirements:

Reviewers' comments:

Reviewer's Responses to Questions

**Comments to the Author**

1. If the authors have adequately addressed your comments raised in a previous round of review and you feel that this manuscript is now acceptable for publication, you may indicate that here to bypass the “Comments to the Author” section, enter your conflict of interest statement in the “Confidential to Editor” section, and submit your "Accept" recommendation.

Reviewer #2: All comments have been addressed

Reviewer #3: All comments have been addressed

2. Is the manuscript technically sound, and do the data support the conclusions?

Reviewer #2: Yes

Reviewer #3: Yes

3. Has the statistical analysis been performed appropriately and rigorously? 

Reviewer #2: Yes

Reviewer #3: Yes

4. Have the authors made all data underlying the findings in their manuscript fully available?

Reviewer #2: Yes

Reviewer #3: Yes

5. Is the manuscript presented in an intelligible fashion and written in standard English?

Reviewer #2: Yes

Reviewer #3: Yes

6. Review Comments to the Author

Reviewer #2: I am satisfied with how the authors have addressed my previous comments. I think that this study serves as the basis for a more complete comprehension of EHEC pathogenesis after exposure to antibiotics. I look forward to seeing more studies that investigate the impact of antibiotics on EHEC virulence and host tissue damage.

Reviewer #3: The manuscript submitted by Kijewski and cols. describe a comparative functional genomic analysis of Escherichia coli O157:H7 under stress condition mediated by exposition to ciprofloxacin. The functional analysis is based in transcriptomic and proteomic techniques. The findings reported by the authors showed the differential expression of different genes and proteins related both virulence and physiological processes that might contribute to pathogenesis of this pathogen. These results are of interest to specialists in the field of physiology, virulence and pathogenicity bacteria.

Major concerns

The authors describe that proteins were collected at 12 hours. However, the growth curve lacks this point.

The experimental design is not clear. In the methodology section, the authors describe that the samples corresponding to the 2h time were evaluated by transcriptomic analysis, and samples corresponding to 3h and 12h were analyzed by different proteomic systems. It’s not clear why each sample is evaluated by a different approach/technique. Why are all samples not analyzed by the same system or technique? I do not understand the use of two LC-MS systems to perform proteomic analysis. Why were 3 of the 12h samples analyzed by two different systems? Proteomic analysis evaluates the functional genome at protein level and transcriptomic at transcriptional level, several events or factors influence in the genic expression. So, to better integrate or correlate the transcriptomic and proteomic dataset, all samples must be analyzed by both transcriptomics and proteomics.

I think that the topic “Factors associated with hypervirulence” is not an appropriate term. Hyper Enterohemorrhagic escherichia coli is subdivided into eight clades; stains related to clade 6 and 8 are reported as more virulent. EDL933 belongs to clade 3. The authors describe several unknown function proteins or hypothetical proteins, phage proteins as factors associated with hypervirulence. The presence of these genes/proteins that are inserted into phage sequence or plasmid is not a parameter to consider a gene or protein associated with hypervirulence, once experimental study has not been performed. Moreover, motility genes are not considered as hypervirulent genes of pathogenic E. coli. StcE and EspP and other genes/proteins from pO157 are described as virulence factors and are highly conserved in E. coli O157:H7.

Minor concerns

Line 118: change “fig. 2” to “Fig. 2”

Line 130: I don’t understand as a p > 0.05 is considered statistically significant.

Table 4: change “Adhesion” to “Adherence factors”

Table 4 is related to adhesion factors. However, the authors report the “Type VI secretion system tip protein VgrG” in this table. The VgrG is an effector molecule that plays a role in the aggregation of actin molecules and alteration of host cell morphology. Please remove this protein from table 4.

Line 518: “...involves a range of different metabolic processes…”, however, only LPS biosynthesis is reported in the Results and Discussion.

Line 557: change “over night” to “overnight"

7. PLOS authors have the option to publish the peer review history of their article (what does this mean?). If published, this will include your full peer review and any attached files.

Reviewer #2: No

Reviewer #3: No

---

## [Author Response · Author response to Decision Letter 1]

11 Dec 2023

Response to Reviewers

Thank you for your thoughtful feedback and suggestions to our manuscript. We appreciate the time and effort you've dedicated to reviewing our work. We hope you find our revised manuscript satisfactory.

Major concerns

The authors describe that proteins were collected at 12 hours. However, the growth curve lacks this point.

Response to reviewer:

Unfortunately, during the time of conducting the experiments, we did not measure growth at the 12-hour time point. Subsequently, we regret to inform you that following our faculty's relocation to a new building, we no longer have access to a BSL3 laboratory for working with EHEC. Despite our efforts to explore collaboration with other institutions, we have been unable to secure access during the afternoon/night hours required for the experiment.

While we are unable to address this specific concern at present, we want to assure you that we have carefully considered your feedback. 

The experimental design is not clear. In the methodology section, the authors describe that the samples corresponding to the 2h time were evaluated by transcriptomic analysis, and samples corresponding to 3h and 12h were analyzed by different proteomic systems. It’s not clear why each sample is evaluated by a different approach/technique. Why are all samples not analyzed by the same system or technique? I do not understand the use of two LC-MS systems to perform proteomic analysis. Why were 3 of the 12h samples analyzed by two different systems? 

Response to reviewer: 

We completely agree with the reviewer that the best option for analyzing proteomics samples would be on the same systems. However, and unfortunately- two events prohibited us from doing this in the present project. The first limitation was the corona pandemic that greatly inhibited our access to equipment and later, when access again was available- machines that had been idle for longer periods were no longer functioning properly. Second, there were (as always) limited funds in the projects so we had to prioritize and therefore rerunning previously analyzed samples was not an option. However, as we explain more thoroughly in the next point, as we analyzed the three samples on both systems, the data become both valid and comparable. 

Proteomic analysis evaluates the functional genome at protein level and transcriptomic at transcriptional level, several events or factors influence in the genic expression. So, to better integrate or correlate the transcriptomic and proteomic dataset, all samples must be analyzed by both transcriptomics and proteomics.

Response to reviewer: 

We also think the reviewer addresses a very interesting point in proteomics- how to compare samples analyzed by different systems. With an increasing amount of different MS based data being publicly available for re-analysis, there is perhaps a discussion to be had on how to incorporate and utilize that data with newer technology that are, and will be, far better than the technology used to generate the deposited data. 

In our work, even with access to even better equipment after the pandemic, we chose to work on systems that were more closely related than the current generation of IM-MS. We hope the reviewer agrees with us that even though not perfect (as we could not perform all analysis on the same system), we managed to utilize and extract more data from “older” systems in a sensible manner by following the reasons below. There will always be a limitation when performing LFQ by the amount of MS1 peaks (precursors) that are positively identified as peptides (on MS2 level) that can subsequently be used for quantitation by the software (sw). Commonly, the peak picking algorithm (ppa) of the sw assign peaks (precursor) that are quantifiable, within biological replicates based on the positive identification of a precursor (MS1 peak) in a single (or if specified- multiple) sample. i.e., MS1 peaks that are not positively identified by MS2 in a single analysis are still used for quantitation if the same peak is identified in a biological replicate. We selected a single sample from each of the 3x 12h groups and analyzed it with a Qexactive (Qex) system, whereas all other samples were analyzed on an Orbitrap XL (XL). These 3 samples were only analyzed on the Qex system- not on both systems. Although the Qex is a major upgrade from the XL in multiple areas, the major difference is in the drastically increased amount of MS2 events that the Qex exhibits that leads to a drastic increase in peptide identifications. We often had to build exclusion or inclusion lists in the XL system to fragment all precursors that were present in samples and build comprehensive protein identification lists. Therefore, from a set of biological replicates that should contain mostly the same peptides as MS1 precursors- by identifying a lot of precursors in a single sample and utilize that as a template for the other biological replicate samples within the same group we would increase the amount of good (and true) quantitative data from the peak picking algorithm rather than using imputation. The reviewer is correct that our wording in the method description was a unclear and to describe it more clearly we have changed the text page 33 line 666 from “All 12 h samples were analyzed on an Ultimate 3000 nano-HPLC (Dionex, Sunnyvale, CA, USA) coupled to an LTQ-Orbitrap XL (OXL) mass spectrometer (MS) (ThermoElectron, Bremen, Germany).” to “4 (of 5) of the 12 h samples were analyzed on an Ultimate 3000 nano-HPLC (Dionex, Sunnyvale, CA, USA) coupled to an LTQ-Orbitrap XL (OXL) mass spectrometer (MS) (ThermoElectron, Bremen, Germany).”” And the text page 33 line 669 “Whereas all 3 h samples and 3 of the 12 h samples (1 from each sample group) were analyzed using an Ultimate 3000 nano-UHPLC system (Dionex, Sunnyvale, CA, USA) connected to a QExactive (QEx) MS (ThermoElectron, Bremen, Germany) equipped with a nano electrospray ion source” to “Whereas all 3 h samples and 3 of the 12 h samples (the final sample from each sample group) were analyzed using an Ultimate 3000 nano-UHPLC system (Dionex, Sunnyvale, CA, USA) connected to a QExactive (QEx) MS (ThermoElectron, Bremen, Germany) equipped with a nano electrospray ion source”

I think that the topic “Factors associated with hypervirulence” is not an appropriate term. Hyper Enterohemorrhagic escherichia coli is subdivided into eight clades; stains related to clade 6 and 8 are reported as more virulent. EDL933 belongs to clade 3. The authors describe several unknown function proteins or hypothetical proteins, phage proteins as factors associated with hypervirulence. The presence of these genes/proteins that are inserted into phage sequence or plasmid is not a parameter to consider a gene or protein associated with hypervirulence, once experimental study has not been performed. Moreover, motility genes are not considered as hypervirulent genes of pathogenic E. coli. StcE and EspP and other genes/proteins from pO157 are described as virulence factors and are highly conserved in E. coli O157:H7.

Response to reviewer:

We think we must partly and respectfully disagree with it not being an appropriate term. We have changed the “factors” with “genes and proteins” as it is more precise. As you state, motility genes as a whole, is not considered hypervirulent genes, and when the word “factor” is used, the reader can misunderstand and think that we mean that for example motility is connected to hypervirulence, when we only mean certain motility related proteins/genes. We thank the reviewer for that. However, as we have written in the manuscript, Amigo et al 2016, identified many proteins that were overexpressed in the hypervirulent clades 6 and 8. We think that it is appropriate to term these overexpressed proteins and genes as “associated with hypervirulence”, as the overexpression in clade 6 and 8 compared to EDL933 becomes defining for these clades. When we treated the non-hypervirulent strain EDL933 with ciprofloxacin, from clade 3, many of these genes and proteins that were overexpressed in clade 6 and 8 was highly upregulated. Since we do explain this in the paper, we think this warrants calling them “genes and proteins associated with hypervirulence”, as “genes and proteins identified as overexpressed in clade 6 and 8 of EHEC O157:H7 by Amigo et al 2016” is too long and tedious.

Minor concerns

Line 118: change “fig. 2” to “Fig. 2”

Response to reviewer:

This has now been corrected.

Line 130: I don’t understand as a p > 0.05 is considered statistically significant.

Response to reviewer: 

Thank you so much for pointing out that we had placed the “smaller than or equal to” the wrong way, this was an oversight, and has now been changed. 

Table 4: change “Adhesion” to “Adherence factors”

Response to reviewer:

Thank you, this has been changed in the revised manuscript.

Table 4 is related to adhesion factors. However, the authors report the “Type VI secretion system tip protein VgrG” in this table. The VgrG is an effector molecule that plays a role in the aggregation of actin molecules and alteration of host cell morphology. Please remove this protein from table 4.

Response to reviewer: Thank you for pointing that out, it has now been removed.

Line 518: “...involves a range of different metabolic processes…”, however, only LPS biosynthesis is reported in the Results and Discussion.

Answer for reviewer: Both figure 4 and 5 show changes in genes and proteins in many metabolic processes. Since the focus of this paper is on changes in virulence related genes and proteins, the metabolic processes are not addressed and discussed in detail, but they are noted. For example, in line 161-162 and in lines 169-172.

Line 557: change “over night” to “overnight" 

 Answer for reviewer: 

Changed

---

## [Decision Letter · Decision Letter 2]

19 Jan 2024

PONE-D-23-24337R2Transcriptomic and proteomic analysis of the virulence inducing effect of ciprofloxacin on enterohemorrhagic Escherichia coliPLOS ONE

Dear Dr. Aspholm,

Thank you for submitting your manuscript to PLOS ONE. After careful consideration, we feel that it has merit but does not fully meet PLOS ONE’s publication criteria as it currently stands. Therefore, we invite you to submit a revised version of the manuscript that addresses the points raised during the review process.

We look forward to receiving your revised manuscript.

Kind regards,

Fernando Navarro-Garcia

Academic Editor

PLOS ONE

Journal Requirements:

Reviewers' comments:

Reviewer's Responses to Questions

**Comments to the Author**

1. If the authors have adequately addressed your comments raised in a previous round of review and you feel that this manuscript is now acceptable for publication, you may indicate that here to bypass the “Comments to the Author” section, enter your conflict of interest statement in the “Confidential to Editor” section, and submit your "Accept" recommendation.

Reviewer #3: (No Response)

2. Is the manuscript technically sound, and do the data support the conclusions?

Reviewer #3: Partly

3. Has the statistical analysis been performed appropriately and rigorously? 

Reviewer #3: Yes

4. Have the authors made all data underlying the findings in their manuscript fully available?

Reviewer #3: Yes

5. Is the manuscript presented in an intelligible fashion and written in standard English?

Reviewer #3: Yes

6. Review Comments to the Author

Reviewer #3: In the study performed by Amigo et al. (2016), the authors describe a group of proteins overexpressed in strains with a phenotype more virulent than EDL933. Amigo et al. do not name these proteins as associated with hypervirulence. The overexpression of these proteins in more virulent strains does not necessarily mean that these proteins are associated with their increased virulence; once comparative experimental studies were not carried out to evaluate the true role of these proteins in the virulence of each strain. Therefore, the term: “genes and proteins associated with hypervirulence” is inappropriate. I recommend the authors remove the complete topic “Increased expression of genes and proteins associated with hypervirulence” because there is no experimental evidence that this set of proteins is responsible for more virulent phenotypes of E. coli O157:H7 strains.

7. PLOS authors have the option to publish the peer review history of their article (what does this mean?). If published, this will include your full peer review and any attached files.

Reviewer #3: No

---

## [Author Response · Author response to Decision Letter 2]

26 Jan 2024

Dear reviewers, we are thankful for your thorough review of our manuscript. You considered the term: “genes and proteins associated with hypervirulence” inappropriate and recommended us to remove the complete topic “Increased expression of genes and proteins associated with hypervirulence” because of lacking experimental evidence that this set of proteins is responsible for more virulent phenotypes of E. coli O157:H7 strains. 

This topic has now been removed. Some of the genes defined as “hypervirulence-associated” in the previous version of the manuscript are phage associated, associated with motility or carried by the pO157 virulence plasmid. Therefore, we have moved some of the text from the previous “hypervirulence” topic to the topics that address these groups of genes/proteins (but without claiming an association with hypervirulence). Additionally, we've incorporated minor revisions into the manuscript to correct spelling errors and improve the wording.

We hope that your concern has been fully addressed in the revised version of the manuscript.

---

## [Editor Report · Decision Letter 3]

30 Jan 2024

Transcriptomic and proteomic analysis of the virulence inducing effect of ciprofloxacin on enterohemorrhagic Escherichia coli

PONE-D-23-24337R3

Dear Dr. Aspholm,

We’re pleased to inform you that your manuscript has been judged scientifically suitable for publication and will be formally accepted for publication once it meets all outstanding technical requirements.

Kind regards,

Fernando Navarro-Garcia

Academic Editor

PLOS ONE